# Improving Reasoning for Diffusion Language Models via Group Diffusion Policy Optimization

**Kevin Rojas**[1]*, **Jiahe Lin**[2], **Kashif Rasul**[2], **Anderson Schneider**[2]
**Yuriy Nevmyvaka**[2], **Molei Tao**[1†], **Wei Deng**[2†]
[1]Georgia Institute of Technology   [2]ML Research, Morgan Stanley
{kevin.rojas,mtao}@gatech.edu, weideng056@gmail.com

## Abstract

Diffusion language models (DLMs) enable parallel, order-agnostic generation with iterative refinement, offering a flexible alternative to autoregressive large language models (LLMs). However, adapting reinforcement learning (RL) fine-tuning to DLMs remains an open challenge because of the intractable likelihood. Pioneering work such as diffu-GRPO (Zhao et al., 2025) estimated token-level likelihoods via one-step unmasking. While computationally efficient, this approach is severely biased. A more principled foundation lies in sequence-level likelihoods, where the evidence lower bound (ELBO) serves as a surrogate. Yet, despite this clean mathematical connection, ELBO-based methods have seen limited adoption due to the prohibitive cost of likelihood evaluation. In this work, we revisit ELBO estimation and disentangle its sources of variance. This decomposition motivates reducing variance through fast, deterministic integral approximations along a few pivotal dimensions. Building on this insight, we introduce **Group Diffusion Policy Optimization (GDPO)**, a new RL algorithm tailored for DLMs. GDPO leverages simple yet effective *Semi-deterministic Monte Carlo* schemes to mitigate the variance explosion of ELBO estimators under vanilla double Monte Carlo sampling, yielding a provably lower-variance estimator under tight evaluation budgets. Empirically, GDPO achieves consistent gains over pretrained checkpoints and outperforms diffu-GRPO, one of the state-of-the-art baselines, on the majority of math, reasoning, and coding benchmarks.

## 1 Introduction

Large language models (LLMs) (Radford et al., 2018; 2019; Brown et al., 2020; Achiam et al., 2023; Dubey et al., 2024; Team, 2025) have have revolutionized modern science by providing exceptionally general-purpose representations and abstractions. Their training typically proceeds in

---

*Kevin did this work while interning at Morgan Stanley. † Correspondence.

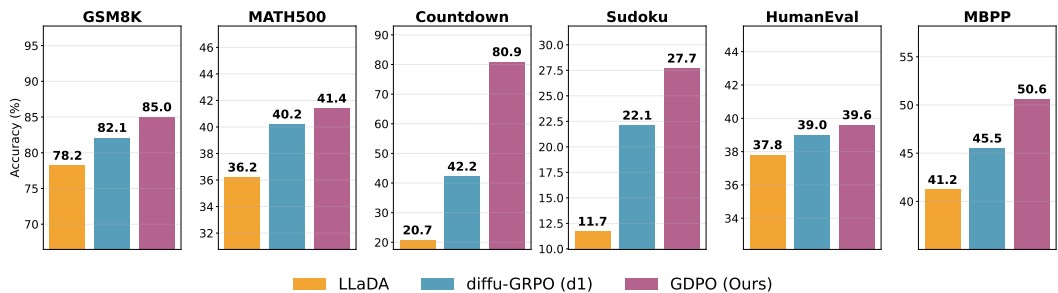

Figure 1: Across reasoning, planning, and coding tasks, our GDPO algorithm for diffusion language models — *using the best of 128/256/512 generations* — significantly outperforms the LLaDA baseline and prior RL methods such as diffu-GRPO.

two stages: a pretraining stage, where vast corpora are used to optimize the next-token prediction objective and endow the model with broad world knowledge and linguistic representations; and a post-training stage, where alignment techniques adapt the raw model outputs for improved reliability and usability (Bai et al., 2022). Among alignment methods, reinforcement learning (RL) (OpenAI, 2024) has emerged as especially promising for post-tuning LLMs on complex tasks with well-defined reward signals. DeepSeekMath (Shao et al., 2024) advances this direction by scaling and stabilizing RL through Group Relative Policy Optimization (GRPO), showing that RL can reduce the computational burden of alignment (Guo et al., 2025; Team et al., 2025) while substantially improving LLMs on reasoning-intensive tasks such as mathematics (Cobbe et al., 2021; Hendrycks et al., 2021a), planning (Ye et al., 2025a; Liu et al., 2025a), coding (Austin et al., 2021b), and multimodality (Venkatraman et al., 2024). Nevertheless, key limitations remain: inference is slow, generation is constrained to a left-to-right order, and early mistakes cannot be revised, often leading to error propagation.

By contrast, discrete diffusion models (Campbell et al., 2022; Lou et al., 2024; Shi et al., 2024; Gat et al., 2024; Sahoo et al., 2024; Nie et al., 2025; Ye et al., 2025b) — often referred to as Diffusion Language Models (DLMs) — offer greater flexibility and versatility. DLMs support faster inference (Arriola et al., 2025; Khanna et al., 2025; Song et al., 2025), iterative refinement through token revisions and remasking (Wang et al., 2025), knowledge transfer via adaptation from autoregressive models (Gong et al., 2025a), and demonstrate superior performance to autoregressive LLMs in low-data regimes (Prabhudesai et al., 2025). These advantages have motivated growing efforts to adapt RL methods originally developed for LLMs to DLMs (Zhao et al., 2025; Zhu et al., 2025; Gong et al., 2025b; Zekri & Boullé, 2025; Shankar, 2025; Huang et al., 2025). However, the absence of a straightforward autoregressive structure complicates likelihood estimation at both the token and sequence levels — quantities that many RL objectives fundamentally rely on.

Previous works have extended GRPO (Zhao et al., 2025; Gong et al., 2025b;b; Tang et al., 2025) and considered different heuristics to approximate the token-level likelihood; such methods were computationally scalable, but their connection to the mathematical foundations of DLMs remain unclear. A different approach considered extending DPO (Zhu et al., 2025), however, their method required many network evaluations, which limits its applicability.

In this paper, we introduce Group Diffusion Policy Optimization (GDPO), a novel RL algorithm built to enhance reasoning for diffusion language models (DLMs). We analyze the variance decomposition of the sequence-level ELBO, which clarifies why prior double Monte Carlo estimators lead to a large variance issue and tend to be computationally expensive. Motivated by these findings, we propose fast yet effective integral approximation strategies based on a *Semi-deterministic Monte Carlo* scheme. This approach enables GDPO to solve complex reasoning tasks for DLMs while provably reducing variance under tight evaluation budgets. Empirically, GDPO consistently improves upon pretrained checkpoints and outperforms state-of-the-art baselines, including diffu-GRPO (Zhao et al., 2025), across diverse math, reasoning, and coding benchmarks.

## 2 PRELIMINARIES

### 2.1 MASKED DIFFUSION LANGUAGE MODELS (MDMs)

Diffusion models (Ho et al., 2020; Song et al., 2021) have demonstrated remarkable success in continuous domains such as image synthesis, but their extension to discrete spaces like text remains less explored. To address this gap, discrete diffusion models (Austin et al., 2021a; Lou et al., 2024; Shi et al., 2024; Sahoo et al., 2024)—often referred to as Diffusion Language Models (DLMs)—introduce masking noise to progressively corrupt sequences and are trained to model the marginal distribution of the induced reverse dynamics. In what follows, we formally define the forward and reverse processes along with the training objectives that characterize DLMs.

**Forward process.** Given a clean sequence data $y_0 \sim \pi_{\text{data}}$ and timestamp $t \in [0, 1]$, the forward process $y_t \sim \pi_{t|0}(\cdot|y_0)$ factorizes as

$$\pi_{t|0}(y_t|y_0) = \prod_{i=1}^{L} \pi_{t|0}(y_t^i|y_0^i), \quad \pi_{t|0}(y_t^i|y_0^i) = \text{Cat}((1-t)e_{y_0^i} + te_M),$$

where $e_{y_0^i}$ is the one-hot vector that encodes the position of token $y_0^i$ and $e_M$ is the one-hot vector for the mask token $M$; and $\mathrm{Cat}(\cdot)$ denotes the categorical distribution. Thus, each coordinate $y_t^i$ is independently replaced by the mask token with probability $t$ and otherwise remains unchanged.

**Reverse process.** The reverse process aims to reconstruct the original sequence $y_0$ from a corrupted sequence $y_t$. Given a probabilistic prediction of the a token $y_\theta^i$, we can write down the transition for any $s < t$ as:

$$q_{s|t}(y_s|y_t) = \prod_{i=1}^{L} q_{s|t}(y_s^i|y_t, y_\theta^i), \quad q_{s|t}(y_s^i|y_t, y_\theta^i) = \begin{cases} \mathrm{Cat}(e_{y^i}), & y_t^i \neq M, \\ \mathrm{Cat}\left(\frac{s}{t}e_M + \frac{t-s}{t}y_\theta^i\right), & y_t^i = M. \end{cases}$$

**Denoising objective.** With the linear noise schedule and time-independent conditional probabilities (Ou et al., 2025), the reverse transition is often approximated by $\pi_\theta(y_0^i|y_t)$, trained via a simple training objective (Zhu et al., 2025):

$$-\mathbb{E}_{y_0 \sim p_{\mathrm{data}}}\mathbb{E}_{t \sim \mathcal{U}[0,1]}\mathbb{E}_{y_t \sim \pi_t(\cdot|y_0)}\left[\frac{1}{t}\sum_{i=1}^{L}\mathbf{1}[y_t^i = M]\log \pi_\theta(y_0^i|y_t)\right]. \tag{1}$$

Notably, the loss function of diffusion language models provides a lower bound for the likelihood known as evidence lower bound (ELBO):

$$\mathcal{L}_{\mathrm{ELBO}}(y|q) = \mathbb{E}_{t \sim \mathcal{U}[0,1]}\mathbb{E}_{y_t \sim \pi(\cdot|y)}\left[\frac{1}{t}\sum_{i=1}^{L}\mathbf{1}[y_t^i = M]\log \pi_\theta(y^i|y_t, q)\right] \leq \log \pi(y|q) \tag{2}$$

where $q$ usually denotes a prompt and $y$ is its answer. DLMs are conceptually similar to BERT (Devlin et al., 2019) in that both rely on token masking to train language representations. However, they differ in that BERT masks a fixed proportion of tokens and predicts them in a single step, while DLMs adopt a time-varying masking schedule and iteratively denoise from full corruption, thus yielding a true generative model.

## 2.2 REINFORCEMENT LEARNING WITHOUT VALUE NETWORKS

Policy gradients (Williams, 1992) have become the workhorse for post-training large language models (LLMs). Among them, proximal policy optimization (PPO) (Schulman et al., 2017) remains the most widely used. However, PPO's reliance on a value network for advantage estimation inevitably increases both computational cost and training instability.

**Group Relative Policy Optimization (GRPO)** (Shao et al., 2024) addresses this limitation by eliminating the value network. Instead, it leverages a Monte Carlo (MC) estimator constructed from multiple sampled answers. For a given prompt $q$ and candidate answers $\{y_g\}_{i=1}^{G}$, the GRPO objective is:

$$\mathcal{L}^{\mathrm{GRPO}}(\theta) = \mathbb{E}_x \mathbb{E}_{y_g \sim \pi_{\mathrm{old}}}\left[\frac{1}{G}\sum_{g=1}^{G}\frac{1}{|y_g|}\sum_{i=1}^{|y_g|}\min\left(r_g^i A_g, \mathrm{clip}(r_g^i, 1-\epsilon, 1+\epsilon)A_g\right) - \beta \mathrm{KL}(\pi_\theta||\pi_{\mathrm{ref}})\right],$$

where the importance ratio and normalized advantage are defined as:

$$r_g^i(y) = \frac{\pi_\theta(y_g^i|q, y_g^{<i})}{\pi_{\mathrm{old}}(y_g^i|q, y_g^{<i})}, \quad A_g = \frac{R_g - \mathrm{mean}(R_1, \ldots, R_G)}{\mathrm{std}(R_1, \ldots, R_G)}, \tag{3}$$

with $R_g = R(q, y_g)$ denoting the sequence-level reward. Notably, although likelihoods are defined at the token level, rewards are assigned only at the sequence level.

Despite their wide adoption, LLMs face key limits: slow inference, rigid left-to-right generation, and error propagation. DLMs mitigate these with parallel, iterative refinement, but their flexibility complicates likelihood estimation and challenges RL-based post-training.

**Diffu-GRPO:** A pioneering effort in fine-tuning diffusion language models was made by Zhao et al. (2025), who proposed *Diffu-GRPO*, an adaptation of GRPO for masked diffusion. They approximated the sequence-level likelihood via a fast but coarse mean-field network evaluation, and

introduced a practical scheme for this approximation. Specifically, their method starts by perturbing the input prompt $q$ with random noise to obtain $q'$. They then consider fully masked sequence $q \oplus M \oplus \cdots \oplus M$, where $\oplus$ denotes concatenation and $M$ is the masked token, they perform a single-step unmasking to estimate

$$\log p_\theta(y_g^i|q' \oplus M \oplus \cdots \oplus M).$$

This approach has two key virtues: it yields likelihood estimates for every token, and it does so with only one forward pass of the network, ensuring computational efficiency. Empirically, Diffu-GRPO achieves consistent performance gains across a wide range of tasks. However, the one-step unmasking in the mean-field manner introduces significant bias: since tokens are generated sequentially, important token correlations are discarded.

## 3 IMPROVING REASONING VIA GROUP DIFFUSION POLICY OPTIMIZATION

GRPO improves computational efficiency and training stability by estimating advantages from group statistics rather than training a value network. In autoregressive LLMs, its effectiveness relies on two factors: (1) accurate sequence likelihoods, naturally supported by the left-to-right factorization, and (2) token-level importance ratios. In Diffusion Language Models (DLMs), however, the order-agnostic generation paradigm (Ou et al., 2025; Kim et al., 2025) renders both sequence likelihoods and token-level ratios intractable.

### 3.1 REVISITING SEQUENCE-LEVEL LIKELIHOOD FOR DLMS

To address these challenges, much of the field has focused on fast but coarse token-level approximations. Token-level methods leverage per-token probabilities to provide fine-grained control over model updates and act as a stabilizing force during training. To make token-level training feasible despite intractable likelihoods, Zhao et al. (2025) introduced heuristic mean-field approximations. Although efficient and empirically effective, these methods fail to capture sequential dependencies and often overweight individual tokens. Gong et al. (2025b) later improved the approximation by incorporating two complementary *random* timesteps, yet a general solution remains elusive.

In contrast, sequence-level objectives provide more faithful training signals but are difficult to apply in DLMs, since the order-agnostic generation paradigm precludes exact likelihood evaluation (Ou et al., 2025). We review the evidence lower bound (ELBO) as a surrogate of the likelihood:

$$\mathbb{E}_{t\sim\mathcal{U}[0,1]}\mathbb{E}_{y_t\sim\pi_t(\cdot|y)}\left[\frac{1}{t}\sum_{i=1}^{L}\mathbf{1}[y_t^i = M]\log\pi_\theta(y^i|y_t,q)\right] \leq \log\pi(y|q),$$

which provides a principled avenue for extending sequence-level RL methods to DLMs. However, the computational cost of this substitution remains unclear.

**Variance–Cost Dilemma**. Although more principled, sequence-level objectives face a fundamental trade-off: accurate likelihood estimates demand expensive network evaluations, while cheaper approximations suffer from high variance or bias. For instance, Nie et al. (2025) report needing up to 128 samples for reliable estimates, incurring prohibitive cost; Zhu et al. (2025) reduces this to eight evaluations, yet the overhead remains substantial, and a systematic understanding of the variance is still lacking. This tension highlights a central challenge:

*Designing estimators that are both efficient and low-variance remains an open problem.*

### 3.2 DISENTANGLING VARIANCE IN ELBO

To tackle the variance–cost dilemma, we begin by analyzing the different sources of variance in approximating the ELBO. From Eq.(2), two distinct sources of randomness emerge: **(1) Random Time:** sampling $t$, which determines the overall masking level; and **(2) Random Masking:** selecting which tokens are masked given that ratio, introducing additional variance. Our analysis disentangles the contribution of each source to the variance of the loss function.

To investigate this we leverage 1000 different prompts pulled from the OpenWeb dataset. Figure 2 reports the mean and variance of the loss as functions of time, along with the percentage of variance

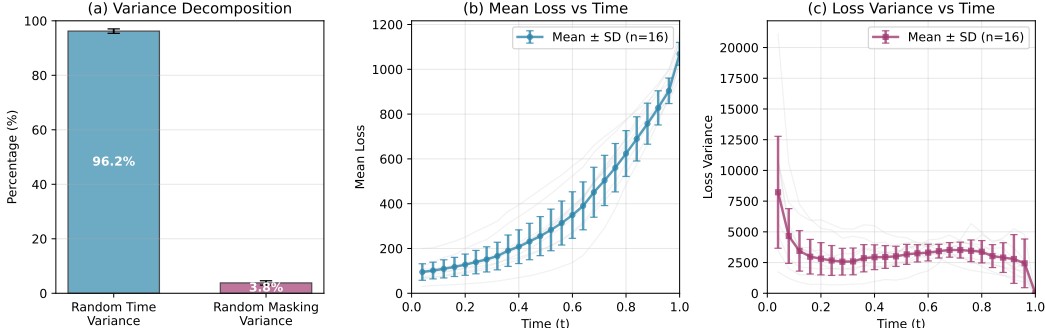

Figure 2: We plot the mean and variance of the loss functions as a function of the noise level $t$. (a) We observe that most of the variance comes from picking the random time (b) The loss function follows a simple, predictable shape across many prompts. (c) The loss variance varies highly at the end but stabilizes for most times.

attributed to each source. A detailed derivation can be found in Appendix A. Several key observations are given next.

**Random time dominates the variance:** As shown in Figure 2(a), the majority of variance arises from sampling the timestamps that control masking ratios. This is intuitive: varying the ratio drastically changes the input—ranging from nearly unmasked to fully masked—which produces large disparities in the loss and inflates variance.

**The loss curve exhibits a simple structure:** In Figure 2(b), the loss as a function of time reveals a smooth, clear structure. Approximating the ELBO boils down to computing the area under this curve, indicating that the problem is naturally suited for *deterministic* integral approximation to suppress the variance.

**Variance across timesteps is stable:** As shown in Figure 2(c), variance peaks near $t = 0$, stabilizes across intermediate masking ratios, and decays to zero as inputs become fully masked. This aligns with observations from Zhu et al. (2025) and indicates that only a small number of samples are needed for the inner integral.

### 3.3 VARIANCE REDUCTION VIA SEMI-DETERMINISTIC MONTE CARLO

To achieve low-variance estimates under tight evaluation budgets, we limit naive Monte Carlo sampling and adopt *deterministic* integration methods to avoid the slow MC convergence of $O(N^{-1/2})$.

**Deterministic time:** Motivated by the observation in Figure 2(a), instead of considering the problem as a double Monte Carlo problem, we consider it to be a time integral to eliminate the large variance caused by random time:

$$\mathcal{L}_{\text{ELBO}}(y|q) = \int_0^1 \mathbb{E}_{y_t \sim \pi_t(\cdot|y)} \left[ \frac{1}{t} \sum_{i=1}^{L} \mathbf{1}[y_t^i = M] \log \pi_\theta(y^i|y_t, q) \right] \mathrm{d}t \leq \log \pi(y|q). \quad (4)$$

**Numerical quadrature:** We further approximate this integral using a standard quadrature with $N$ points, then our estimate is of the form:

$$\mathcal{L}_{\text{ELBO}}(y|q) \approx \sum_{n=1}^{N} w_n \underbrace{\sum_{k=1}^{K} \left[ \frac{1}{t_n} \sum_{i=1}^{L} \mathbf{1}[(y_{t_n}^{[k]})^i = M] \log \pi_\theta(y^i|y_{t_n}^{[k]}, q) \right]}_{l(\pi_\theta; y, q, t_n)}. \quad (5)$$

where $y_{t_n}^{[k]} \sim \pi_{t_n}(\cdot|y)$ and $\{w_n\}_{n=1}^{N}$ are the associated weights and the inner expectation is approximated using Monte-Carlo estimates.

Due to the deterministic–stochastic nature in the integration, we refer to it as a **Semi-deterministic Monte Carlo (SDMC)** scheme.

This representation serves several advantages, firstly, it is guaranteed to approximate the ELBO when $N$ and $K$ are large enough. Secondly, by fixing the time points in the first integral we significantly reduce the variance of our estimator. Thirdly, as observed in Figure 2 (b), where the loss function is plotted for several different prompts, the loss function has a simple shape, specifically, it is strictly increasing and convex, which makes it well-suited for integral approximations via quadratures. Furthermore as observed in Figure 2 (c) the variance remains relatively constant across many noise levels, which allows for stable computations and results.

In practice, we opt for utilizing a simple and effective Gaussian Quadratures, which are known to have fast convergence rates (Dahlquist & Björck, 2008). Furthermore, based on our variance analysis we employ a single Monte-Carlo estimate for the inner integral. Consequently, the total number of network evaluations per likelihood computation matches exactly the number of quadrature points $N$.

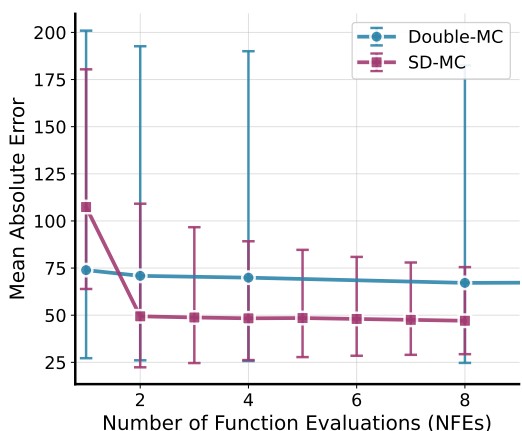

Figure 3: Estimation error and variance for Double Monte Carlo vs our Semi-deterministic Monte Carlo method. SD-MC achieves lower bias and variance, with most benefits obtained using only 2–3 points.

To evaluate our method in a controlled setting, we draw $1000$ samples from the OpenWeb dataset and estimate sequence likelihoods using both Monte Carlo and Gaussian quadrature with varying numbers of function evaluations. For each sample, we repeat the estimation 16 times and measure both the deviation from a ground-truth approximation (computed with $1024$ Monte Carlo samples) and the variance of the estimator. As shown in Figure 3, quadrature-based estimators consistently exhibit lower bias and variance, yielding accurate estimates of the ELBO. Notably, most of the gains are achieved with as few as 2 or 3 quadrature points, which we adopt in practice.

With the flexibility to reconcile efficiency and statistical accuracy, we are ready to introduce **Group Diffusion Policy Optimization (GDPO)**, a policy gradient method tailored to diffusion language models with group statistics. Compared with Diffu-GRPO (Zhao et al., 2025), which employs efficient but crude estimates of token likelihoods, GDPO uses sequence-level likelihoods, which are made efficient through the SDMC scheme. This reformulation of the importance weights from the token-level to the sequence-level has the added benefit of preserving the semantics of the advantage estimates, and the ELBO-based objective fits naturally within the discrete diffusion framework while retaining the merits of GRPO-style updates. Formally, the GDPO loss is defined as:

$$\mathcal{L}^{\text{GDPO}}(\theta) = \mathbb{E}_x \mathbb{E}_{y_g \sim \pi_{\text{old}}} \left[ \frac{1}{G} \sum_{g=1}^{G} \frac{1}{|y_g|} \min\left(r_g A_g, \text{clip}(r_g, 1-\epsilon, 1+\epsilon) A_g\right) - \beta \text{KL}(\pi_\theta || \pi_{\text{ref}}) \right],$$
(6)

where the importance weights and advantage estimates are both done at the *sequence* level:

$$r_g(x) = \frac{\exp(\mathcal{L}_{\text{ELBO}}(y_g|x))}{\exp(\mathcal{L}_{\text{ELBO}}^{\text{old}}(y_g|x))}, \quad A_g = R_g - \text{mean}(R_1, \dots, R_G).$$

Here $\mathcal{L}_{\text{ELBO}}^{\text{old}}$ represents the ELBO evaluated under the old policy and $R_g = R(q, y_g)$, and we utilize unnormalized advantage estimates to avoid the bias (Liu et al., 2025b). The use of the ELBO is in spirit similar to GSPO (Zheng et al., 2025).

### 3.4 OVERVIEW OF THEORETICAL RESULTS

We provide a brief theoretical analysis for the proposed ELBO estimate that is based on a *Semi-deterministic Monte Carlo*, with an emphasis on its asymptotic error bounds when the total number of Monte Carlo samples $K$ and the number of integration points $N$ become large. Note that a

---

**Algorithm 1** GDPO: Group Diffusion Policy Optimization for diffusion language models (DLMs), with ELBO estimated via a Semi-deterministic Monte Carlo (SDMC) scheme.

---

**Require:** Reference model $\pi_{\text{ref}}$, distribution over prompts $\mathcal{D}$, completions per prompt $G$, inner updates $\mu$, quadrature points and weights $\{(t_n, w_n)\}_{n=1}^{N}$
1: Initialize from a reference model $\pi_\theta \leftarrow \pi_{\text{ref}}$
2: **while** not converged **do**
3:     $\pi_{\theta_{\text{old}}} \leftarrow \pi_\theta$
4:     Draw a prompt sample $q \sim \mathcal{D}$
5:     Generate $G$ completions $y_g \sim \pi_{\theta_{\text{old}}}(\cdot \mid q), \ g \in [G]$
6:     Estimate reward $r_g$ and advantage $A_g^k(\pi_{\theta_{\text{old}}})$ using Eq.(3) for each $y_g$.
7:     **for** $n = 1, \ldots, \mu$ **do**                       ▷ For each gradient update
8:         **for** $g = 1, \ldots, G$ **do**
9:             $\mathcal{L}_{\text{ELBO}}(y_g|q) \leftarrow 0$
10:             **for** $n = 1, \ldots, N$ **do**             ▷ Evaluate quadrature over time
11:                 $\mathcal{L}_{\text{ELBO}}(y_g|q) \leftarrow \mathcal{L}_{\text{ELBO}}(y_g|q) \ + \ w_n \cdot \ell(\pi_\theta; y_g, q, t_n)$
12:             **end for**
13:         **end for**
14:         Evaluate GDPO objective in Eq.(6) using $\{\mathcal{L}_{\text{ELBO}}(y_g|q)\}_{g=1}^{G}$.
15:         Optimize $\pi_\theta$ via AdamW.
16:     **end for**
17: **end while**
18: **return** $\pi_\theta$

---

"classic" alternative to this estimator would be a double Monte Carlo one, and its error bound (in the form of MSE) would scale at $O(\frac{1}{NK})$. For our proposed estimator, the analysis points to the following results:

- **decomposition of the MSE**: the mean squared error of the estimator can be decomposed into the sum of Monte Carlo variance and the square of the integration bias, which resembles the classic variance-bias decomposition of statistical estimators;

- **rates under very general conditions**: the variance term scales as $O(\frac{1}{NK})$; with a generic integration scheme (e.g., the Riemann sum [1]), the squared integration bias scales as $O(\frac{1}{N^2})$;

- **faster rate with additional assumptions on the log-likelihood**: with an $N$-dependent decay condition on its variance, the variance term would scale faster at the rate of $O(\frac{1}{N^2K})$;

- **quadrature rule for integration can make the bias practically negligible**: when the integrand is twice continuously differentiable and therefore one can use quadrature to perform integration, the squared integration bias would scale either at the rate of $O(\frac{1}{N^4})$ or $O(\frac{1}{N^8})$, depending on the exact integration scheme used. As such, this term becomes practically negligible and the variance term becomes the dominating one.

The upshot is that under certain regularity assumptions of the log-likelihood, the proposed SDMC estimator can attain a faster rate than a generic double Monte Carlo estimator due to the *deterministic* integration. All details are deferred to Appendix B.

Table 1: Asymptotic Error Bounds in relation to Integration Points $N$ and Monte Carlo Samples $K$.

| Setting | Variance | Bias$^2$ |
|---|---|---|
| General conditions / Riemann sum | $O(1/NK)$ | $O(1/N^2)$ |
| Additional assumption & smoothness / Quadrature | $O(1/N^2K)$ | $O(1/N^4)$ or $O(1/N^8)$ |

As noted above, to perform integration using quadrature, the integrand needs to be *sufficiently smooth*. To that end, properties of the integrand are further investigated. In particular, by relating the integrand in Eq.(5) to a KL form, it can be shown that such an KL form is indefinitely

---

[1]In the case of Riemann sum, the integration is approximated as $\int_0^1 g(t)\mathrm{d}t \approx \sum_{n=1}^{N} g(c_n)w_n$, where $c_n$ is any value within the $n$th interval; $w_n \equiv \frac{1}{N}$ when the intervals are equally spaced.

differentiable (namely, living in $\mathcal{C}^\infty$). Further, under some additional assumptions on the likelihood ratio associated with the data distribution and the forward diffusion process, one can prove that the integrand is convex and monotone in $t$. Such conditions can be understood conceptually as follows: masking more tokens corresponds to removing more information, and the penalty grows as masking increases. The implication of these results are two-fold: (1) given the structure of the integrand, the standard quadrature rules are well-suited for this problem; combined with results from the error bound analysis, this justifies the observation that our ELBO estimator exhibits faster convergence and lower variance than the double Monte Carlo one; and (2) the convex shape empirically observed in Figure 2(b) supports the theoretical claim under the additional assumptions, albeit verifying these assumptions can be non-trivial. Details for this part of the results are in Appendix C.

## 4    EXPERIMENTS

We conduct a comprehensive set of experiments. As our base model we use LLaDA-8B-Instruct Zhu et al. (2025) which is a open sourced DLM that has been tuned to follow instructions, but no specific post-training. We investigate the effect of applying GDPO with our SDMC estimator.

**Tasks:** We conduct experiments on mathematical reasoning, planning and coding benchmarks. For (1) Mathematical reasoning: we use the GSM8K (Cobbe et al., 2021) which contains grade school math problems, and MATH500 (Lightman et al., 2023), containing 500 problems drawn from the MATH dataset (Hendrycks et al., 2021b). (2) Planning: this includes two tasks: 4x4 Sudoku puzzles, and Countdown in which given 3 numbers and a target the model must use arithmetic operations to reach the target. (3) Coding: we use the HumanEval (Chen et al., 2021), a benchmark consisting of 164 manually designed Python algorithmic challenges and sanitized MBPP (Austin et al., 2021b), which contains 257 crowd-sourced Python programming tasks.

Table 2: **Model performance on Mathematics and Planning Benchmarks** based on $N = 3$ quadrature points. Green is the best performing model.

| Model | GSM8K | | | MATH500 | | | Countdown | | | Sudoku | | |
|---|---|---|---|---|---|---|---|---|---|---|---|---|
| | 128 | 256 | 512 | 128 | 256 | 512 | 128 | 256 | 512 | 128 | 256 | 512 |
| LLaDA-8B-Instruct | 68.7 | 76.7 | 78.2 | 26.0 | 32.4 | 36.2 | 20.7 | 19.5 | 16.0 | 11.7 | 6.7 | 5.5 |
| + *diffu*-GRPO | 72.6 | 79.8 | 81.9 | 33.2 | 37.2 | 39.2 | 33.2 | 31.3 | 37.1 | 18.4 | 12.9 | 11.0 |
| +SFT+*diffu*-GRPO | 73.2 | 81.1 | 82.1 | **33.8** | 38.6 | 40.2 | 34.8 | 32.0 | 42.2 | 22.1 | 16.7 | 9.5 |
| +wD1 | N/A | 80.8 | 82.3 | N/A | 34.4 | 39.0 | N/A | 51.2 | 46.1 | N/A | 25.2 | 24.2 |
| + SFT + GDPO | **78.39** | **82.79** | **84.99** | 33.2 | **39.6** | **41.4** | **64.45** | **74.22** | **80.86** | **27.69** | **26.42** | **26.17** |

### 4.1    MAIN RESULTS

**GDPO consistently improves the checkpoint and outperforms diffu-GRPO** Table 2 demonstrates the performance of the baseline model, as well as other token level approaches. Our results demonstrate that we can better improve the baseline than the token-based algorithms.

**GDPO outperforms most baselines w/ SFT.** As shown in Table 2, GDPO with 3 quadrature points surpasses existing RL baselines after a few epochs of supervised fine-tuning (SFT), highlighting its strong ability to enhance reasoning performance.

**GDPO enhances reasoning in coding tasks.** We fine-tune the model on the KodCode-Light-RL-10K dataset (Xu et al., 2025), which spans a wide range of coding problems at varying difficulty levels, each validated through unit tests. We found that $N = 3$ led to slightly better results than $N = 2$, and we report GDPO with $N = 3$ in Table 3. The results demonstrate that GDPO with

Table 3: Model performance on Coding with $N = 3$ quadrature points. Green is best.

| Model | HumanEval | | | MBPP | | |
|---|---|---|---|---|---|---|
| | 128 | 256 | 512 | 128 | 256 | 512 |
| LLaDA-8B-Instruct | 27.4 | 35.5 | 37.8 | 36.2 | 41.2 | 40.4 |
| + *diffu*-GRPO | **29.3** | 39.0 | 34.8 | 42.0 | 45.5 | 41.6 |
| + GDPO | 26.2 | **39.6** | **39.0** | **43.6** | **50.6** | **47.1** |

3 quadrature points consistently improves performance across most baselines. Most strikingly, on the MBPP benchmark, RL fine-tuning achieves a substantial *10% accuracy gain* over the pretrained model without SFT.

**GDPO improves performance beyond the training sequence length.** Building on observations by Zhao et al. (2025) that Diffu-GRPO enhances generalization to longer contexts, we find that GDPO achieves this effect to an even greater extent. On 512-token sequences, GDPO consistently outperforms all baselines while largely preserving performance at shorter lengths. We attribute this to the use of sequence-level likelihoods, which promote more uniform improvements across token positions, in contrast to token-level methods that retain generation-order biases, as noted by Gong et al. (2025b).

**GDPO is computationally efficient** Notably we are able to obtain these remarkable results training on only 2 H100 GPUs, although training can be accelerated with 8 H100 GPUs. This is remarkable as practitioners often have a limited computational budget and GDPO opens the opportunities to such needs.

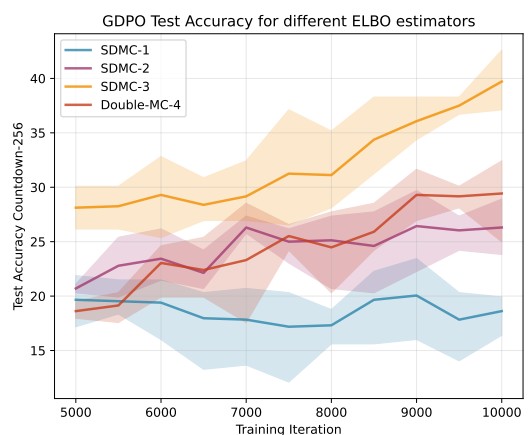

Figure 4: Test accuracy with different training iterations and ELBO estimators on the Countdown dataset.

**Importance of the ELBO approximation:** To demonstrate that accurately approximating the likelihood is of vital importance we fix ourselves to the Countdown dataset. We pick this one as we observed the greatest increases in performances in such dataset which allows for easier interpretation of the results. We train 4 models using the same set of hyperparameters, only varying the ELBO approximation method. We evaluate the test accuracy as a function of the training iteration every 500 iterations.

As observed in Figure 4, estimators that are more accurate result in better improvements on the RL pipelines. Furthermore, the sheer number of function evaluations is not enough to guarantee good results. For instance, SDMC-3 can significantly outperform the naive Monte Carlo estimator even when it uses more evaluations. This demonstrates that accurately designing the estimator is of vital importance in GDPO.

## 5 CONCLUSION

We proposed Group Diffusion Policy Optimization (GDPO), an RL algorithm designed specifically for diffusion language models (DLMs). By leveraging sequence-level likelihoods through the evidence lower bound (ELBO), our approach disentangles and mitigates the major sources of variance that hinder prior methods. In particular, we replace the inefficient double Monte Carlo estimation with a simple and fast *Semi-deterministic Monte Carlo* sampling, yielding a provably lower-variance and more computationally efficient estimator. Extensive experiments demonstrate that GDPO consistently improves over pretrained checkpoints and surpasses strong baselines such as diffu-GRPO across math, reasoning, and coding benchmarks. We expect even stronger performance with more powerful pretrained checkpoints, though this remains outside the scope of the present study. Taken together, these findings highlight GDPO as both a theoretically principled and practically effective paradigm for aligning DLMs. We believe that the use of *Semi-deterministic Monte Carlo* sampling offers a simple and viable path to handling the large variance issue in the sequence-level ELBO estimation of DLMs, and future work can explore more effective *deterministic–stochastic integration* schemes with *data-driven* quadrature weights and locations to minimize the large variance further.

### ACKNOWLEDGMENTS

We are thankful to CoreWeave, whose purpose-built AI cloud platform powered all of our experiments. KR and MT gratefully acknowledge the partial support of NSF Grant DMS-2513699, DOE Grants NA0004261, SC0026274, and Richard Duke Fellowship.

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

## Supplementary Material for "*Improving Reasoning for Diffusion Language Models via Group Diffusion Policy Optimization*"

In this supplementary material, we present additional details on variance decomposition in §A, provide a theoretical analysis of asymptotic error bounds in §B, justify key properties of the integrand in §C, describe the experimental setup in §D, a comparison against Gong et al. (2025b) in §E and include representative generated samples in §F.

## A   MORE DETAILS ON VARIANCE DECOMPOSITION

We start by studying the different sources of variance in approximating the ELBO, in the case of masked diffusion models the equation is given by:

$$\mathbb{E}_t \mathbb{E}_{y_t \sim \pi(\cdot|y)} \left[ \frac{1}{t} \sum_{i=1}^{L} \mathbf{1}[y_t^i = M] \log \pi_\theta(y^i|y_t, q) \right] \leq \log \pi(y|q) \tag{7}$$

From looking at this expression we observe that there are two sources of variance, **(1) Time Sampling:** we must sample $t$ which determines the noise level, and **(2) Random Masking:** we must sample $y_t$ which injects noise to the clean $x_0$. We start by analyzing how each source of randomness affects the variance of an estimator. To simplify our discussion, we will rewrite the integrand in a simplified way:

$$\mathbb{E}_t \mathbb{E}_{y_t \sim \pi(x_t|y)} [Z(t, x_t)] \leq \log \pi(y|q), \tag{8}$$

where $Z(t, y_t) := \frac{1}{t} \sum_{i=1}^{L} \mathbf{1}[y_t^i = M] \log \pi_\theta(y^i|y_t, q)$ is a random variable w.r.t. the joint distribution of $(t, y_t)$. We are interested in studying $\text{Var}(Z(t, y_t))$ and how it decomposes with respect to $t, y_t$, by the law of total variance we can decompose this as:

$$\text{Var}(Z(t, y_t)) = \underbrace{\mathbb{E}_t[\text{Var}(Z \mid t)]}_{\text{Var given by } Z|t} + \underbrace{\text{Var}_t(\mathbb{E}[Z \mid t])}_{\text{Var given by } t}. \tag{9}$$

The above expression indicates that we can understand variance as the sum of two components, where each term corresponds to a distinct source of variance. We argue that most of the variance comes from selecting the noise level $t$. When the noise level is $t = 0$ we range from fully unmasked to fully masked, creating a large disparity in the noise samples and subsequently causing large swings in the variance.

To test this we evaluate each term in 9 for 1000 different prompts pulled from the OpenWeb dataset. We plot the mean and variance as function of time in Figure 2. This figure reveals very important facts:

1. Most of the variance is coming from randomly selecting the timestamps/ noise levels and not from injecting noise into the text.

2. The loss function when observed as a function of time yields a simple structure.

For this reason, we advocate for *fixed timestamps* for likelihood approximation and embracing a *Semi-deterministic Monte Carlo sampler* instead of the naïve double Monte Carlo approximations.

## B   ANALYSIS OF ASYMPTOTIC ERROR BOUNDS

In this section, we analyze the error bound of approximating the ELBO via the proposed Semi-Deterministic Monte Carlo scheme, focusing on the training objective given in Eq.(2), namely,

$$\mathcal{L}_{\text{ELBO}}(y|q) = \mathbb{E}_{t \sim \mathcal{U}[0,1]} \mathbb{E}_{y_t \sim \pi_t(\cdot|y)} \left[ \frac{1}{t} \sum_{i=1}^{L} \mathbf{1}[y_t^i = M] \log \pi_\theta(y^i|y_t, q) \right].$$

**Notation and definition.** We proceed by first defining the relevant quantities that will be used in the ensuing technical analysis. Define

$$Z_t := \frac{1}{t} \sum_{i=1}^{L} \mathbf{1}[y_t^i = M] \log \pi_\theta(y^i | y_t, q),$$

which is a *random variable* with respect to the distribution of $y_t$, where $y_t \sim \pi_t(\cdot | y)$. Let

$$g(t) := \mathbb{E}_{y_t \sim \pi_t(\cdot | y)}(Z_t), \tag{10}$$

and the Monte Carlo approximation of $g(t)$ be $g_{\mathrm{MC}}(t) := \frac{1}{K} \sum_{k=1}^{K} (Z_t^{[k]})$, where $Z_t^{[k]}$ are iid draws of $Z_t$. For convenience, we let $\mathcal{I}(y_t) := \{i \in \{1, \cdots, L\} : y_t^i = M\}$ and $c_t = |\mathcal{I}(y_t)|$, the cardinality of $\mathcal{I}(y_t)$. Note that $Z_t$ can be rewritten as $Z_t \equiv \frac{1}{t} \sum_{i \in \mathcal{I}(t)} \log p_\theta(y^i | y_t, q)$.

With these definitions, the training objective of interest is given by

$$\mathcal{L}_{\mathrm{ELBO}}(y|q) = \mathbb{E}_{t \sim \mathcal{U}[0,1]}(g(t)) = \int_0^1 g(t) \mathrm{d}t, \tag{11}$$

and its approximation via a Semi-deterministic Monte Carlo scheme (namely, Eq.(5)) can be written as

$$\widehat{\mathcal{L}}_{\mathrm{ELBO}}(y|q) := \sum_{n=1}^{N} w_n g_{\mathrm{MC}}(t_n), \tag{12}$$

which is a numerical approximation of the integral in the Monte Carlo approximation $\widetilde{\mathcal{L}}_{\mathrm{ELBO}}(y|q) := \int_0^1 g_{\mathrm{MC}} \mathrm{d}t$ at points $t_n, n = 1, \cdots, N$. For the time being, we assume a generic numerical approximation of the integral, and using the notation involving $w_n$.

The approximation error of Eq.(12) is given by

$$\mathbb{E}\left( \left[ \widehat{\mathcal{L}}_{\mathrm{ELBO}} - \mathcal{L}_{\mathrm{ELBO}} \right]^2 \right) = \mathbb{E}\left( \left[ \sum_{n=1}^{N} w_n g_{\mathrm{MC}}(t_n) - \int_0^1 g(t) \mathrm{d}t \right]^2 \right) \tag{13}$$

$$= \underbrace{\mathbb{E}\left[ \left( \sum_{n=1}^{N} w_n g_{\mathrm{MC}}(t_n) - \sum_{n=1}^{N} w_n g(t_n) \right)^2 \right]}_{\text{MC variance}} + \underbrace{\left( \sum_{n=1}^{N} w_n g(t_n) - \int_0^1 g(t) dt \right)^2}_{\text{integration bias}^2};$$

this holds due to the independence of Monte Carlo samples, since

$$\mathbb{E}\left[ \sum_{n=1}^{N} w_n g_{\mathrm{MC}}(t_n) \right] = \sum_{n=1}^{N} w_n \mathbb{E}[g_{\mathrm{MC}}(t_n)] = \sum_{n=1}^{N} w_n g(t_n),$$

and therefore the second term boils down to the integration bias. It can be seen that this becomes the classical mean-squared-error of an estimator, which can be decomposed into the variance (from a total of $K$ Monte Carlo samples) and the bias$^2$ (from the approximation of the integral) at $N$ points $t_1, \cdots, t_N$. The bounds of these two terms will be analyzed separately in the sequel.

## B.1 THE VARIANCE TERM

To analyze this term, we start with the *pointwise* Monte Carlo error.

**Lemma B.1.** *Assume the following conditions hold:*

C1. $\mathbb{E}_{x_t}(c_t) < \infty, \quad \mathbb{E}_{x_t}(c_t^2) < \infty$;

C2. $\exists \delta > 0 : \delta < \pi_\theta(y^i | y_t, q) < 1 - \delta$, *for all $x_0^i$, $x_t$.*

*Then, the following holds*

$$\mathbb{E}\left( g(t) - g_{MC}(t) \right)^2 = \tfrac{1}{K} \sigma^2(t), \qquad \text{where} \quad \sigma^2(t) := Var(Z_t). \tag{14}$$

*Proof.* To establish the results in Eq.(14), the following assumptions need to hold:

    A1. Measurability: $Z_t$ is jointly measurable in $t$ and $y_t$;

    A2. Integrability: $\mathbb{E}(|Z_t|) < \infty$, $\forall\, t \in [0, 1]$;

    A3. Finite variance: $\sigma^2(t) < \infty$, $\forall\, t \in [0, 1]$.

By writing $Z_t$ as $Z_t = \frac{1}{t} \sum_{i \in \mathcal{I}(y_t)} \log \pi_\theta(y^i \mid y_t, q)$, it can be verified that C1 and C2 are sufficient for A1-A3 to hold. $\square$

**Remark B.1.** *Interpretation of conditions C1 and C2 are given as follows: C1 requires the number of masked tokens to be controlled, and C2 requires the conditional likelihood to be well behaved.*

Coming back to the variance term, by Lemma B.1, at each time point $t$ (i.e., $t_n$'s), the Monte Carlo estimate $g_{\mathrm{MC}}(t_n)$ introduces random error whose pointwise variance is given by $\frac{1}{K}\sigma^2(t_n)$. Consequently, the variance of the estimator $\widehat{\mathcal{L}}_{\mathrm{ELBO}}$ is given by

$$\mathbb{E}\left[\left(\sum_{n=1}^{N} w_n g_{\mathrm{MC}}(t_n) - \sum_{n=1}^{N} w_n g(t_n)\right)^2\right] = \mathbb{E}\left[\left(\sum_{n=1}^{N} w_n \underbrace{\left(g(t_n) - g_{\mathrm{MC}}(t_n)\right)}_{:=u_n}\right)^2\right]$$

$$= \mathbb{E}\Big(\sum_{n=1}^{N} w_n u_n\Big)^2 = \frac{1}{K}\Big(\sum_{n=1}^{N} w_n^2 \sigma^2(t_n)\Big); \qquad (15)$$

the last equality holds since for $t_n \neq t_n$, due to the independence of MC samples across $n$, we have that $\mathrm{Cov}(u_n, u_m) = 0$ for $n \neq m$; further note that $\mathbb{E}(u_n) = 0$ and $\mathbb{E}u_n^2 = \frac{1}{K}\sigma^2(t_n)$.

Proposition B.1 shows that under a refinement of condition C1, Eq.(15) is bounded and therefore its rate can be derived accordingly.

**Proposition B.1.** *Suppose the following condition holds for $c_t$ (defined identically to that in Lemma B.1):*

    *C1'. $\mathbb{E}_{x_t}(c_t) \leq C_0 t^2$ for some constant $C_0 > 0$.*

*Assume also that $w_n = O(1/N)$ and $t_n$ are approximately equally spaced. Then, the following holds:*

$$\frac{1}{K}\sum_{n=1}^{N} w_n^2 \sigma^2(t_n) \leq \frac{C_0}{K}\sum_{n=1}^{N} w_n^2 t_n = O\Big(\frac{1}{K} \cdot \frac{1}{N^2}\sum_{n=1}^{N} t_n\Big) = O\Big(\frac{1}{K}\frac{1}{N^2}N\int_0^1 t\mathrm{d}t\Big) = O\Big(\frac{1}{KN}\Big).$$
$$(16)$$

**Remark B.2.** *Condition C1' guarantees that $\int_0^1 \sigma^2(t) < \infty$, which then implies the finiteness of $\sum_{n=1}^{N} w_n^2 \sigma^2(t_n)$ as $N \to \infty$, provided that the weights $w_n$ corresponds to a valid integration scheme; i.e., $w_n > 0$, $\sum_{n=1}^{N} w_n = 1$ and the mesh of the points $t_n$ becomes finer as $N \to \infty$. Empirically, C1' can be operationalized by considering a masking scheme where token $x_t^i$ is masked independently across $i = 1, \cdots, L$ with probability $t$, and $c_t \sim \mathrm{Bin}(L, t)$. Finally, the assumption on weights $w_n = O(\frac{1}{N})$ and that $t_n$'s being roughly equally spaced are both fairly reasonable for all integration schemes (simple Riemann, midpoint, trapezoid, Simpson).*

The next proposition shows that when an $N$-dependent decay condition is satisfied by the log-likelihood, then a rate faster than the one established in Eq.(16) can be obtained.

**Proposition B.2.** *Suppose the following condition holds*

    *C1". $\mathbb{E}_{x_t}(c_t^2) \leq C_{token}$ for some positive constant $C_{token}$.*

*In addition, assume the log-likelihood satisfies the following, for some positive constant $C_{log\text{-}lik}$:*

$$Var\big(\log \pi_\theta(y^i | y_t, q)\big) \leq C_{log\text{-}lik}\frac{t^2}{N}.$$

*Then, the following holds*

$$\frac{1}{K} \sum_{n=1}^{N} w_n^2 \sigma^2(t_n) = O\left(\frac{1}{KN^2}\right). \tag{17}$$

*Proof.* Given the definition of $\sigma^2(t)$ (see, e.g., Eq.(14)), under condition C1", one has

$$\sigma^2(t) \leq \frac{C_{\text{token}}^2}{t^2} \text{Var}\big( \log \pi_\theta(y^i|y_t, q) \big).$$

Then, for weights $\max_n w_n = O(\frac{1}{N})$ (which are the weights for all standard integration rules) and $t_n = \frac{n}{N}$, we get

$$\sigma^2(t_n) \leq \frac{C_{\text{token}}^2}{t_n^2} C_{\text{log-lik}} \frac{t_n^2}{N} = C_0 \frac{1}{N}, \quad \text{where } C_0 = C_{\text{token}}^2 C_{\text{log-lik}}.$$

Hence, we have a uniform bound on $\sigma^2(t_n) \leq C_0 \frac{1}{N}$ for every $n = 1, \cdots, N$. Using the above uniform bound, we obtain

$$\sum_{n=1}^{N} w_n^2 \sigma^2(t_n) \leq \frac{C_0}{N} \sum_{n=1}^{N} w_n^2 = \frac{C_0}{N} \left[ N \frac{1}{N^2} \right] = \frac{C_0}{N^2}.$$

Consequently, it follows that

$$\frac{1}{K} \sum_{n=1}^{N} w_n^2 \sigma^2(t_n) = O\left(\frac{1}{KN^2}\right).$$

$\square$

Some intuition behind the $N$-dependent decay condition is given as follows. Specifically, for the log-likelihood to satisfy such a condition, one compatible specification is given by

$$\log \pi_\theta(y^i|y_t, q) = h(t) + \frac{t}{\sqrt{N}} \epsilon_t, \quad \text{where} \quad \mathbb{E}(\epsilon_t) = 0, \ \text{Var}(\epsilon_t) \leq \sigma_0 \text{ for some } \sigma_0 > 0; \quad \forall t;$$

$h(t)$ is a smooth Lipschitz function. Under this specification, the fluctuation of the likelihood is small for small $t$ (little masking) and increases for large $t$.

### B.2 THE BIAS TERM

Next, we analyze the bias term induced by approximating the integral $\int_0^1 g(t)\mathrm{d}t$ via numerical integration schemes.

The next lemma provides the rate of the bias for the case where some quadrature rule is used, under certain assumptions of $g(t)$.

**Lemma B.2.** *Suppose that $g(t) : (0, 1] \to \mathbb{R}$ is a deterministic and continuously twice differentiable function (i.e., in $\mathcal{C}^2$). Further, suppose we use some quadrature rule (midpoint, trapezoidal, Simpson, etc.) to approximate its integral at points $\{t_n\}_{n=1}^{N} \in [0, 1]$; i.e.,*

$$\int_0^1 g(t)dt \approx \sum_{n=1}^{N} w_n g(t_n), \tag{18}$$

*where weights $w_n$'s are chosen according to the specific quadrature rule in use. Then, such an approximation scheme introduces deterministic integration bias, given by*

$$E_N(g) = | \int_0^1 g(t)\mathrm{d}t - \sum_{n=1}^{N} w_n g(t_n) |,$$

*and it scales as follows:*

- *for midpoint or trapezoidal rule, as $O(\frac{1}{N^2})$;*

- *for Simpson, as $\mathcal{O}(\frac{1}{N^4})$.*

*Proof.* These are standard results from numerical analysis; see, e.g., Dahlquist & Björck (2008); Atkinson (2008). □

The above lemma directly yields the following result: for the integration bias[2] that appears in the decomposition of Eq.(13), it follows that for midpoint or trapezoidal rule,

$$\left(\sum_{n=1}^{N} w_n g(t_n) - \int_0^1 g(t)dt\right)^2 = O(\frac{1}{N^4});$$

and for Simpson, the rate is given by $O(\frac{1}{N^8})$.

**Remark B.3.** *The above rates suggest that by applying a quadrature rule—provided that $g(t)$ satisfies certain properties—the integration bias[2] has become practically negligible, relative to the variance term that dominates. However, if one uses a generic integration scheme instead, such as the Riemann sum, the integration bias would scale at the rate of $O(\frac{1}{N})$ and thus the bias[2] at $O(\frac{1}{N^2})$.*

As it can be seen from Remark B.3, the "gain" by considering a Semi-Deterministic Monte Carlo scheme is partly built on the fact that the integration bias[2] can be practically negligible, which hinges on whether one can use quadrature rules for integration. However, to apply such rules, $g(t)$ needs to be continuously twice differentiable, and its properties are further studied in Appendix C.

## C  PROPERTIES OF THE INTEGRAND IN THE NUMERICAL QUADRATURE

For this section, we will denote $p_0$ a probability distribution on $[M]^D$ and denote $p_t$ the $t$ marginal of the forward process under masked diffusion and $\pi_t$ the learned marginal. Our goal is to state some conditions under which the loss function would be well-suited for numerical quadratures. We start by noting that from Proposition 1 of Benton et al. (2024) we can write the ELBO in the following form:

$$\mathcal{L}_{\text{ELBO}}(y|q) = \mathbb{E}_{t\sim\mathcal{U}[0,1]}\mathbb{E}_{y_t\sim\pi(\cdot|y)}\left[\frac{1}{t}\sum_{i=1}^{L}\mathbf{1}[y_t^i = M]\log\pi_\theta(y^i|y_t,q)\right]$$

$$= \int_0^1 \mathbb{E}\left[\Phi\left(\frac{p_t(y)}{\pi_t(y_t)}\right)\right]\mathrm{d}t + C$$

$$= \int_0^1 \frac{\mathrm{d}}{\mathrm{d}t}\mathrm{KL}(p_t||\pi_t)\mathrm{d}t + C,$$

where $\Phi$ is the score matching operator introduced in Benton et al. (2024). This calculation allows to realize that to study the integrand it is enough to study $\mathrm{KL}(p_t||\pi_t) =: f(t)$. Note that with this definition, the integrand $g(t)$ defined in the previous section satisfies $g(t) = f'(t) + C$ for some constant $C$. By Proposition (1.1) in Benton et al. (2024) we know that $\Phi \geq 0$, therefore $f'(t) \geq 0$.

**Notation:** We first introduce the following notation: let $q^{\mathcal{S}}$ denotes the probability distribution $q$ marginalized over the entries not in $\mathcal{S}$. The following lemma states that we can write $f(t)$ as a polynomial with coefficients relating to the average KL under different levels of masking.

**Lemma C.1.** *The* KL *divergence for two distributions over time evolving according to masked diffusion is $\mathcal{C}^\infty$ as a function of $t$ and we can write:*

$$\mathrm{KL}(p_t||\pi_t) = \sum_{k=1}^{D}\sum_{\substack{\mathcal{S}\subset[D]\\|\mathcal{S}|=k}} t^k(1-t)^{D-k}\mathrm{KL}(p_0^{\mathcal{S}}||\pi_0^{\mathcal{S}})$$

$$= \sum_{k=1}^{D} c_k\binom{D}{k}t^k(1-t)^{D-k}$$

*where:*

$$c_k = \frac{1}{\binom{D}{k}} \sum_{\substack{\mathcal{S} \subset [D] \\ |\mathcal{S}| = k}} \mathrm{KL}(p_0^{\mathcal{S}} || \pi_0^{\mathcal{S}})$$

*is the average* KL *divergence over sets of size k.*

*Proof.* The proof relies on fundamental combinatorial facts and the definition of the forward process:

$$\mathrm{KL}(p_t || \pi_t) = \sum_{x \in [M]^D} p_t(x) \log\left(\frac{p_t(x)}{\pi_t(x)}\right)$$

$$= \sum_{k=1}^{D} \sum_{\substack{\mathcal{S} \subset [D] \\ |\mathcal{S}| = k \iff i \in S}} \sum_{\substack{x : x_i = M \\ \iff i \in S}} p_t(x) \log\left(\frac{p_t(x)}{\pi_t(x)}\right)$$

$$= \sum_{k=1}^{D} \sum_{\substack{\mathcal{S} \subset [D] \\ |\mathcal{S}| = k \iff i \in S}} \sum_{\substack{x : x_i = M \\ \iff i \in S}} t^k (1-t)^{D-k} p_0^{\mathcal{S}}(x) \log\left(\frac{t^k(1-t)^{D-k} p_0^{\mathcal{S}}(x)}{t^k(1-t)^{D-k} \pi_0^{\mathcal{S}}(x)}\right)$$

$$= \sum_{k=1}^{D} \sum_{\substack{\mathcal{S} \subset [D] \\ |\mathcal{S}| = k}} t^k (1-t)^{D-k} \sum_{\substack{x : x_i = M \\ \iff i \in S}} p_0^{\mathcal{S}}(x) \log\left(\frac{p_0^{\mathcal{S}}(x)}{\pi_0^{\mathcal{S}}(x)}\right)$$

$$= \sum_{k=1}^{D} \sum_{\substack{\mathcal{S} \subset [D] \\ |\mathcal{S}| = k}} t^k (1-t)^{D-k} \mathrm{KL}(p_0^{\mathcal{S}} || \pi_0^{\mathcal{S}})$$

Using this, we can rewrite the KL as:

$$\mathrm{KL}(p_t || \pi_t) = \sum_{k=1}^{D} \sum_{\substack{\mathcal{S} \subset [D] \\ |\mathcal{S}| = k}} t^k (1-t)^{D-k} \mathrm{KL}(p_0^{\mathcal{S}} || \pi_0^{\mathcal{S}})$$

$$= \sum_{k=1}^{D} c_k \binom{D}{k} t^k (1-t)^{D-k}$$

which gives the result. $\qquad\square$

Given this lemma, it becomes clear that to understand the properties of the integrand, it is enough to understand the following polynomial:

$$f(t) = \sum_{k=1}^{D} c_k \binom{D}{k} t^k (1-t)^{D-k}$$

$$= \sum_{k=1}^{D} c_k B_{k,D}(t)$$

This is a Bernstein polynomial, and its properties have been studied before.

**Forward differences and Bernstein derivatives.** When looking at the derivatives of Bernstein polynomials, the differences of the coefficients appear; to this end denote:

$$\Delta c_k = c_{k+1} - c_k$$

$$\Delta^2 c_k = c_{k+2} - 2c_{k+1} + c_k = (c_{k+2} - c_{k+1}) - (c_{k+1} - c_k) = \Delta_{k+1} - \Delta_k$$
$$\Delta^3 c_k = c_{k+3} - 3c_{k+2} + 3c_{k+1} - c_k = \Delta^2_{k+1} - \Delta^2_k$$

Then it is easy to check that:

**Lemma C.2** (Derivative formulas). *For all $t \in [0, 1]$,*

$$f'(t) = D \sum_{k=0}^{D-1} \Delta c_k \, B_{k,D-1}(t), \tag{19}$$

$$f''(t) = D(D-1) \sum_{k=0}^{D-2} \Delta^2 c_k \, B_{k,D-2}(t), \tag{20}$$

$$f^{(3)}(t) = D(D-1)(D-2) \sum_{k=0}^{D-3} \Delta^3 c_k \, B_{k,D-3}(t). \tag{21}$$

Because $B_{k,n}(t) \geq 0$ on $[0, 1]$, the signs of $f^{(r)}(t)$ are controlled by the signs of the averaged forward differences $\Delta^r c_k$.

**One-step representation via KL chain rule.** Let $F(\mathcal{S}) = \mathrm{KL}(p_0^{\mathcal{S}} \| \pi_0^{\mathcal{S}})$, we are now able to obtain simple, intuitive explanations of the meaning of each coefficient $\Delta_k^i$. For $i \notin \mathcal{S}$, define the (nonnegative) one-step gain

$$\Delta_i(\mathcal{S}) \; := \; F(\mathcal{S} \cup \{i\}) - F(\mathcal{S}) = \mathbb{E}_{X_\mathcal{S} \sim P_0^{(\mathcal{S})}} \big[ \mathrm{KL}\big( P_0(X_i \,|\, X_\mathcal{S}) \,\|\, Q_0(X_i \,|\, X_\mathcal{S}) \big) \big] \geq 0.$$

The above quantity represents the average KL that we obtain by masking an extra token to a given set. Intuitively we expect $\Delta_k$ to be increasing, as masking more tokens reduces the amount of information that we have (This in fact, holds already without extra assumptions). Similarly, we can obtain the iterated differences:

$$\Delta c_k = \mathbb{E}_{\substack{|\mathcal{S}|=k \\ i \notin \mathcal{S}}} \big[ \Delta_i(\mathcal{S}) \big], \tag{22}$$

$$\Delta^2 c_k = \mathbb{E}_{\substack{|\mathcal{S}|=k \\ i,j \notin \mathcal{S}}} \big[ \Delta_i(\mathcal{S} \cup \{j\}) - \Delta_i(\mathcal{S}) \big], \tag{23}$$

$$\Delta^3 c_k = \mathbb{E}_{\substack{|\mathcal{S}|=k \\ i,j,\ell \notin \mathcal{S}}} \big[ \Delta_i(\mathcal{S} \cup \{j, \ell\}) - \Delta_i(\mathcal{S} \cup \{j\}) - \Delta_i(\mathcal{S} \cup \{\ell\}) + \Delta_i(\mathcal{S}) \big]. \tag{24}$$

If such quantities are positive, this will immediately imply that the derivatives in C.2 will be positive, implying important consequences on the loss function. Intuitively, such coefficients being positive correspond to the idea that masking one token when $K$ are masked results in more loss of information than masking a token when $L < K$ are masked. We require these as assumptions and explain the sufficient conditions for them to hold in the following Proposition:

**Proposition C.1** (Sufficient conditions for convexity and increasing integrand). *For all $t \in [0, 1]$:*

1. *(Integrand is positive) For any $P_0, Q_0$, the set function $F$ is monotone: $\mathcal{S} \subseteq \mathcal{T} \Rightarrow F(\mathcal{S}) \leq F(\mathcal{T})$ (data processing under marginalization). Hence $\Delta_i(\mathcal{S}) \geq 0$, $\Delta c_k \geq 0$, and by Eq.(19) we have*

$$\boxed{f'(t) = \frac{\mathrm{d}}{\mathrm{d}t} \mathrm{KL}(P_t \| Q_t) \; \geq \; 0}.$$

2. *(Integrand is increasing in $t$) If the **Increasing Conditional Divergence (ICD)** condition holds:*

$$\Delta_i(\mathcal{S}) \; \leq \; \Delta_i(\mathcal{T}) \qquad \text{for all } \mathcal{S} \subseteq \mathcal{T} \subseteq [D] \setminus \{i\},$$

*then $\Delta^2 c_k \geq 0$ by Eq.(23), and Eq.(20) yields*

$$\boxed{f''(t) = \frac{\mathrm{d}^2}{\mathrm{d}t^2} \mathrm{KL}(P_t \| Q_t) \; \geq \; 0}.$$

3. *(Integrand has convexity in $t$) If, in addition, the **second-order ICD** condition holds:*

$$\left(\Delta_i(\mathcal{S} \cup \{j\}) - \Delta_i(\mathcal{S})\right) \ \leq \ \left(\Delta_i(\mathcal{S} \cup \{j, \ell\}) - \Delta_i(\mathcal{S} \cup \{\ell\})\right)$$

*for all distinct $i, j, \ell$ and $\mathcal{S} \subseteq [D] \setminus \{i, j, \ell\}$, then $\Delta^3 c_k \geq 0$ by Eq.(24), and Eq.(21) gives*

$$\boxed{f^{(3)}(t) = \frac{\mathrm{d}^3}{\mathrm{d}t^3}\mathrm{KL}(P_t \| Q_t) \ \geq \ 0}.$$

# D  EMPIRICAL DETAILS

## D.1  TRAINING DETAILS

We leverage the codebase from Zhao et al. (2025) which in itself leverages the TRL library (von Werra et al., 2020). For our hyperparameters we keep most of the default parameters from Zhao et al. (2025) without any hyperparameter search. We leverage a Low-Rank Adaptation with rank $r = 128$ and scaling factor $\alpha = 64$.

Across all runs we utilize the AdamW optimizer with parameters $\beta_1 = 0.9, \beta_2 = 0.99$ with a weight decay of $0.1$ and gradient clipping at $0.2$. We leverage flash attention and $4-$bit quantization. We found that GDPO usually requires a smaller learning rate than diffu-GRPO and otherwise it can result in diverging models. We save checkpoints every $300$ or $500$ iterations and stop our runs when the reward function has plateaued, for the countdown dataset we observe that the reward function continuous increasing for a very long time, while for the math dataset we observed high-performing checkpoints around 9000 iterations, but report the best one, which happened later in training. We present the learning rate as well as other hyperparameters in Table 4. We present the reward function for the different datasets in Figure 5 and observe that GDPO can offer a steady growth before plateauing.

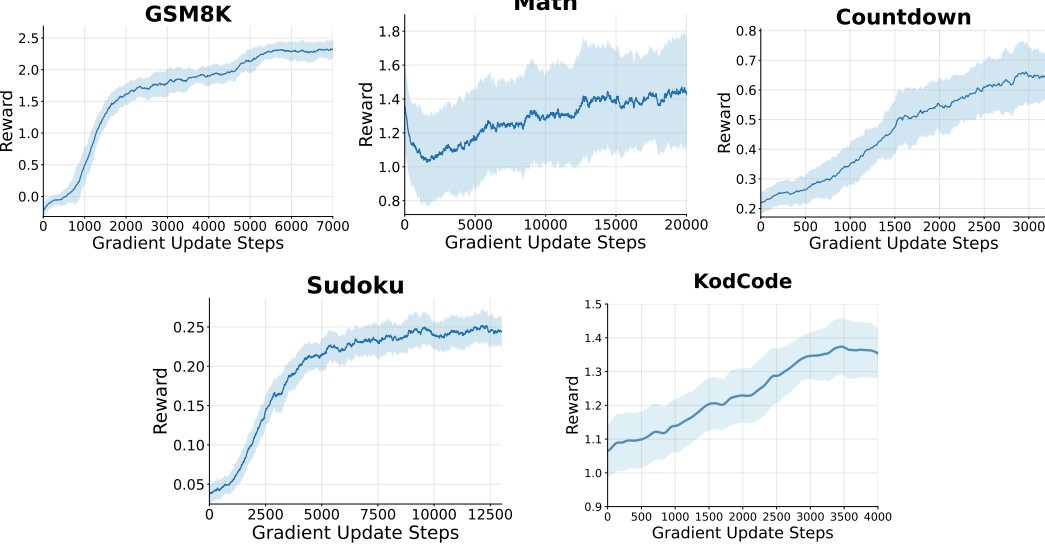

Figure 5: Reward curves during RL training for the models reported in Table 2.

## D.2  REWARD FUNCTIONS

To provide a self-contained exposition, we summarize the reward functions used in RL-based post-training. The design incorporates task-specific incentives that promote formatting fidelity, logical consistency, and solution correctness, with their structure tailored to the demands of each task. This formulation is largely consistent with prior work (Zhao et al., 2025), and our exposition follows established conventions.

**GSM8K** The setup uses a composite reward function composed of several components, including

- **Correctness:** Reward for the parsed answer aligning with the ground truth (+2.0 points).
- **Integer Answer:** Reward for producing a valid integer as the parsed answer (+0.5).
- **Soft Format:** Reward for responses in the pattern `<reasoning>...</reasoning>` `<answer>...</answer>` (+0.5 points).
- **Strict Format:** Reward for outputs matching the strict format and line breaks (+0.5 points)
- **XML Format:** Reward for correct placement of reasoning and answer tags, with +0.125 points per matched pair.

**MATH500**

- **Correctness**: Reward for the parsed answer given in `\boxed{}` and consistent with the ground truth (+2.0).
- **Soft Format**: Reward for the use of `<answer>` tags and `\boxed`, assigned as:
    - +1.00 point when answer tags are included and enclose `\boxed`.
    - +0.75 point when answer tags are included but without `\boxed`.
    - +0.50 point when answer tags are omitted yet `\boxed` is provided.
    - +0.25 point when both answer tags and `\boxed` are missing.

**Countdown**

- **Soft Correctness:** Reward for equations relative to the target value:
    - +1.0 point when the equation reaches the target using only the provided numbers.
    - +0.1 point when the equation does not reach the target but uses the correct numbers.
    - +0 point when both criteria fail.

**Sudoku**

- **Cell Accuracy:** Reward for the fraction of originally empty cells filled correctly.

**Coding** (Gong et al., 2025b)

- **Format:** Reward for generated code adhering to the required Python code-block structure:
    - 1.0 point when the code is enclosed in a valid Python code block ` ```python ``` `, and the snippet parses without syntax errors.
    - 0.5 point when the output is well-formatted but contains Python syntax errors.
    - 0.0 point when the format is invalid or no code block is detected.
- **Code Execution:** Reward for generated code that passes unit tests:
    - Score equals the fraction of test cases passed by executing the code.
    - 0.0 point when the code fails formatting or does not run successfully.

Table 4: **Hyperparameters for different training runs**

| Model | GSM8K | MATH500 | Countdown | Sudoku | KodCode-Light-RL |
|---|---|---|---|---|---|
| Learning Rate | $5e-7$ | $3e-7$ | $1e-6$ | $2e-7$ | $1e-6$ |
| Number of GPUs | 8 | 4 | 8 | 8 | 8 |
| Per-GPU-Batch Size | 6 | 6 | 6 | 6 | 6 |
| Number of Quadrature Points | 3 | 3 | 3 | 3 | 3 |
| Number of iterations of best checkpoint | 6000 | 17000 | 3000 | 11500 | 3800 |

Table 5: **Comparison of training iterations across different configurations**

| Method | Time (Hours) | Number of Iterations | NFEs per Step |
|---|---|---|---|
| diffu-GRPO | 6 | 4500 | 1 |
| GDPO ($N$=2) | 6 | 3800 | 2 |
| GDPO ($N$=3) | 6 | 3500 | 3 |

Table 6: **Comparison of model performance across different configurations.**

| Model | NFE | GSM8K | | | Countdown | | |
|---|---|---|---|---|---|---|---|
| | | 128 | 256 | 512 | 128 | 256 | 512 |
| LLaDA-8B-Instruct | 1 | 68.7 | 76.7 | 78.2 | 36.2 | 41.2 | 40.4 |
| + *diffu*-GRPO | 1 | 73.2 | 81.1 | 82.1 | 34.8 | 32.0 | 42.2 |
| + GDPO ($N$=2) | 2 | **79.08** | **82.94** | 84.61 | 48.83 | 67.19 | 77.34 |
| + GDPO ($N$=3) | 3 | 78.39 | 82.79 | **84.99** | **64.45** | **74.22** | **80.86** |

### D.3 ABLATION STUDIES

**Computational Cost Analysis**  We present an analysis on the additional computational budget introduced by using multiple NFEs per training iteration. To do so, we limit the total training time to a fixed budget of 6 hours. We study this on the countdown dataset and use d1 (Zhao et al., 2025) as our baseline. We use a per-device batch size of 6 and train 3 models using a different number of NFEs per training step. We don't evaluate the test accuracy for this experiment as different methods may require different hyperparameters to achieve optimal performance. Our main goal is to highlight the additional computational overhead introduced by using GDPO with multiple quadrature points.

We present the results of this analysis in Table 5. As expected, increasing the number of network evaluations results in less total training iterations. However, even in the most expensive configuration $N = 3$ we still achieve 77% of the iterations while obtaining the added benefits of GDPO and better likelihood approximation.

**Effect of Quadrature Points**  We further conduct an ablation study in Table 6 to analyze the impact of varying the number of quadrature points when approximating the ground-truth likelihood. Increasing the number of quadrature points consistently improves approximation quality and training stability. Using $N = 2$ already captures most of the gains relative to the ground truth, while $N = 3$ achieves the best overall performance, establishing state-of-the-art results among the evaluated configurations.

## E  COMPARISON WITH DIFFUCODER

One method that appears similar to ours is Gong et al. (2025b) which introduces coupled-GRPO an extension of diffu-GRPO (Zhao et al., 2025) that leverages multiple network evaluations to improve the per-token likelihood evaluations. Diffu-GRPO had originally used the following approximation to the token-level likelihoods given by:

$$\log p_\theta(y^i | q \oplus M \oplus \cdots \oplus M).$$

Coupled-GRPO improves on this by using complementary timestep pairs $(t, \hat{t})$ such that $t + \hat{t} = T$ and complementary masks. Each mask is disjoint and together they cover the entire sequence. Given the two sequences $y_t, y_{\hat{t}}$ one can approximate the **token-level likelihoods** as:

$$\log \pi_\theta(y^i | q) \approx \begin{cases} \log p_\theta(y^i | q \oplus y_t) & \text{if} \quad y_t^i = y^i \\ \log p_\theta(y^i | q \oplus y_{\hat{t}}) & \text{if} \quad y_{\hat{t}}^i = y^i \end{cases}$$

This method has the virtue of evaluating each token under a more realistic setting than the fully masked set up of Zhao et al. (2025). Furthermore, Gong et al. (2025b) showed that it results in variance reduction while remaining computationally efficient.

There are two main differences between both methods. **(1) Token vs Sequence Level probability** GDPO relies on sequence level probabilities, while coupled-GRPO relies on token level likelihoods. Although both methods achieve a variance reduction, it is over a different objective. **(2) Variable number of timesteps** Coupled-GRPO uses 2 sampling times, while GDPO, when implemented through Gaussian Quadratures, can use an arbitrary $N$. Although it should be possible to extend coupled-GRPO to more points, we leave this analysis to future work.

Table 7: **Model performance on Mathematics and Planning Benchmarks** Green is the best performing model.

| Model | Countdown | | | Sudoku | | |
|---|---|---|---|---|---|---|
| | 128 | 256 | 512 | 128 | 256 | 512 |
| LLaDA-8B-Instruct | 20.7 | 19.5 | 16.0 | 11.7 | 6.7 | 5.5 |
| + *diffu*-GRPO | 33.2 | 31.3 | 37.1 | 18.4 | 12.9 | 11.0 |
| +SFT+*diffu*-GRPO | 34.8 | 32.0 | 42.2 | 22.1 | 16.7 | 9.5 |
| +wD1 | N/A | 51.2 | 46.1 | N/A | **25.2** | 24.2 |
| +coupled-GRPO | 33.59 | 43.75 | 47.66 | 23.63 | 21.0 | 21.92 |
| + GDPO | **42.97** | **67.19** | **66.41** | **25.05** | 24.17 | **25.10** |

We perform experiments to analyze the difference in performance between GDPO and coupled-GRPO. We implement diffucoder for the countdown and sudoku benchmarks following the official implementation and present the results in Table 7. We see that couple-GRPO is effective in improving the results when compared to diffu-GRPO but GDPO obtains better results across all tasks.

## F  SAMPLE GENERATIONS

### F.1  GENERATION ON SUDOKU

For the sudoku task we leverage the following prompt and we present samples at 512 tokens.

---

**Sample Generations**

**Prompt**

```
<|startoftext|><|start_header_id|>user<|end_header_id|>
Please solve the following 4x4 Sudoku puzzle. The puzzle is
↪   provided as a 16-character string reading left-to-right,
↪   top-to-bottom, where '0' represents empty cells.
Rules:
- Fill empty cells with digits 1-4
- Each row must contain digits 1-4 exactly once
- Each column must contain digits 1-4 exactly once
- Each 2x2 box must contain digits 1-4 exactly once
Important: Your solution must be a COMPLETE 16-character
↪   string with only the digits 1-4, representing your final
↪   solved grid.
Respond in this exact format:
<reasoning>
Your step-by-step solving process
</reasoning>
<answer>
[16-character solution string with no spaces or separators]
</answer>
```

---

---

**Sample Generations**

**Question**

```
Solve the following Sudoku puzzle: 4320004330100004
```

**Model Generation**

```
To solve the 4x4 Sudoku puzzle, we need to fill in empty
↪  cells with digits 1-4 such that each row, column, and 2x2
↪  box contains all digits exactly once. We's start by
↪  analyzing the given puzzle and filling in the grid step
↪  by step.
1. **Identify the empty cells**:
   - The first row: 43 20
   - The second row: 00 43
   - The third row: 30 10
   - The fourth row: 00 04
2. **Fill the first column**:
   - The first column already has 4, 0, 3, and 0. The only
     ↪  digits left are 1, 2, and 3.
   - Fill the first column: 1 2 3 4
3. **Fill the second column**:
   - The second column already has 3, 4, 0, and 0. The only
     ↪  digits left are 1, 2, and 4.
   - Fill the second column: 1 2 4 4
4. **Fill the third column**:
   - The third column already has 2, 3, 1, and 4. The only
     ↪  digits left are 1, 2, and 3.
   - Fill the third column: 1 2 3 4
5. **Fill the fourth column**:
   - The fourth column already has 0, 0, 0, and 4. The only
     ↪  digits left are 1, 2, and 3.
   - Fill the fourth column: 1 2 3 4
Now, let's fill the 2x2 boxes:
- The first 2x2 box: 1 2 3 4
- The second 2x2 box: 1 2 4 4
- The third 2x2 box: 1 2 3 4
- The fourth 2x2 box: 1 2 3 4
</reasoning>
<>
</>
<answer>
123124
12444
123124
123124
</answer><|eot_id|><|endoftext|>
```

**Ground Truth**

4321124334122134

---

---

**Sample Generations**

**Question**

Solve the following Sudoku puzzle: 3014002020004130

**Model Generation**

```
    To solve the 4x4 Sudoku puzzle, we need to fill in empty
    ↪ cells with digits 1-4 such that each row, column, and 2x2
    ↪ box contains all digits exactly once. We's start by
    ↪ analyzing the given puzzle and filling in the grid step
    ↪ by step.
    1. **Identify the initial digits**:
       - The first row is \"3014\".
       - The second row is \"0020\".
       - The third row is \"2000\".
       - The fourth row is \"4130\".
    2. **Fill the first column**:
       - The first column is \"3014\".
       - The second column is \"0020\".
       - The third column is \"2000\".
       - The fourth column is \"4130\".
    3. **Fill the second column**:
       - The second column is \"0020\".
       - The second column is \"0020\".
       - The third column is \"0020\".
       - The fourth column is \"0130\".
    4. **Fill the third column**:
       - The third column is \"2000\".
       - The third column is \"2000\".
       - The third column is \"2000\".
       - The fourth column is \"0030\".
    5. **Fill the fourth column**:
       - The fourth column is \"4130\".
       - The fourth column is \"4130\".
       - The fourth column is \"4130\".
       - The fourth column is \"0334\".
    Now, let's fill in the 2x2 boxes:
    - The first 2x2 box is \"3014\".
    - The second 2x2 box is \"2020\".
    - The third 2x2 box is \"2000\".
    - The fourth 2x2 box is \"4134\".
    Finally, we can verify that all rows, columns, and boxes are
    ↪ filled correctly according to the Sudoku rules.
    </reasoning>
    <answer>
    12131421221122141334
    </answer><|eot_id|><|endoftext|>
```

**Ground Truth**

3214142323414132

---

---

**Sample Generations**

**Question**

```
Solve the following Sudoku puzzle: 0000100420013142
```

**Model Generation**

```
    To solve the 4x4 Sudoku puzzle, we need to fill in the grid
    ↪  with digits 1-4 such that each row, column, and 2x2 box
    ↪  contains all the digits exactly once. Let's the
    ↪  step-by-step process:
    1. **Identify the missing digits**: Start by identifying the
    ↪  missing digits in the grid. In this case, the missing
    ↪  digits are 1, 2, 3, and 4.
    2. **Analyze the grid**: Look at the grid and identify the
    ↪  positions of the missing digits. The this puzzle, the
    ↪  missing digits are in the first row, first column, and
    ↪  the first 2x2 box.
    3. **Fill in the first row**: The first row must contain the
    ↪  digits 1-4 exactly once. The first row should be 1, 2, 3,
    ↪  4.
    4. **Fill in the first column**: The first column must
    ↪  contain the digits 1-4 exactly once. The first column
    ↪  should be 1, 2, 3, 4.
    5. **Fill in the first 2x2 box**: The first 2x2 box must
    ↪  contain the digits 1-4 exactly once. The first 2x2 box
    ↪  should be 1, 2, 3, 4.
    6. **Fill in the second row**: The second row must contain
    ↪  the digits 1-4 exactly once. The second row should be 2,
    ↪  3, 4, 1.
    7. **Fill in the second column**: The second column must
    ↪  contain the digits 1-4 exactly once. The second column
    ↪  should be 2, 3, 4, 1.
    8. **Fill in the second 2x2 box**: The second 2x2 box must
    ↪  contain the digits 1-4 exactly once. The second 2x2 box
    ↪  should be 2, 3, 4, 1.
    9. **Fill in the third row**: The third row must contain the
    ↪  digits 1-4 exactly once. The third row should be 3, 4, 1,
    ↪  2.
    By following these steps, we can solve the 4x4 Sudoku puzzle.
    </reasoning>
    <answer>
    123412341123413412
    </answer><|eot_id|><|endoftext|>",
```

**Ground Truth**

```
4213132424313142
```

---

## F.2 GSM8K SAMPLES

For the gsm8k task we leverage the following prompt and we present generations at 256 tokens.

### Sample Generations

**Prompt**

```
<|startoftext|><|start_header_id|>user<|end_header_id|>

      You are a math expert. You will be given a question to solve.
      ↪  Solve it step by step. Wrap the final answer in a
      ↪  \\boxed{}.
      Respond in the following format:
      <reasoning>
      Your reasoning here
      </reasoning>
      <answer>
      \\boxed{...}
      </answer>
```

### Sample Generations

**Question**

```
Janet\u2019s ducks lay 16 eggs per day. She eats three for
↪  breakfast every morning and bakes muffins for her friends
↪  every day with four. She sells the remainder at the
↪  farmers' market daily for $2 per fresh duck egg. How much
↪  in dollars does she make every day at the farmers'
↪  market?
```

**Model Generation**

```
      To determine how much Janet makes every day at the farmers'
      ↪  market, we need to follow these steps:

      1. Calculate the total number of eggs laid by Janet's ducks
      ↪  per day.
      2. Subtract the number of eggs she eats for breakfast and
      ↪  bakes for her friends.
      3. Multiply the remaining eggs by the price at which she
      ↪  sells them.

      First, we know that Janet's ducks lay 16 eggs per day.

      Next, we account for the eggs she eats for breakfast and
      ↪  bakes for her friends:
      - She eats 3 eggs for breakfast.
      - She bakes 4 eggs for her friends.

      So, the total number of eggs she uses is:
      \\[ 3 + 4 = 7 \\]

      The number of eggs left to sell is:
      \\[ 16 - 7 = 9 \\]

      Janet sells each egg for $2. Therefore, her daily earnings
      ↪  are:
      \\[ 9 \\times 2 = 18 \\]

      Thus, Janet makes $18 every day at the farmers' market.
      </reasoning>

      <answer>
      \\boxed{18}
      </answer><|eot_id|><|endoftext|>",
```

**Ground Truth**

```
18.0
```

---

### Sample Generations

**Question**

```
Every day, Wendi feeds each of her chickens three cups of
↪   mixed chicken feed, containing seeds, mealworms and
↪   vegetables to help keep them healthy.  She gives the
↪   chickens their feed in three separate meals. In the
↪   morning, she gives her flock of chickens 15 cups of feed.
↪   In the afternoon, she gives her chickens another 25 cups
↪   of feed.  How many cups of feed does she need to give her
↪   chickens in the final meal of the day if the size of
↪   Wendi's flock is 20 chickens?
```

**Model Generation**

```
<|startoftext|><|start_header_id|>user<|end_header_id|>

    You are a math expert. You will be given a question to solve.
    ↪   Solve it step by step. Wrap the final answer in a
    ↪   \\boxed{}.
    Respond in the following format:
    <reasoning>
    Your reasoning here
    </reasoning>
    <answer>
    \\boxed{...}
    </answer>

    Every day, Wendi feeds each of her chickens three cups of
    ↪   mixed chicken feed, containing seeds, mealworms and
    ↪   vegetables to help keep them healthy.  She gives the
    ↪   chickens their feed in three separate meals. In the
    ↪   morning, she gives her flock of chickens 15 cups of feed.
    ↪   In the afternoon, she gives her chickens another 25 cups
    ↪   of feed.  How many cups of feed does she need to give her
    ↪   chickens in the final meal of the day if the size of
    ↪   Wendi's flock is 20 chickens?
    <|eot_id|><|start_header_id|>assistant<|end_header_id|>

    <reasoning>",
    "generations": "
    To determine how many cups of feed Wendi needs to give her
    ↪   chickens in the final meal of the day, we first need to
    ↪   calculate the total amount of feed required for the
    ↪   entire day. Wendi has 20 chickens, and each chicken needs
    ↪   3 cups of feed per day. Therefore, the total amount of
    ↪   feed needed for the day is:
    \\[ 20 \\text{ chickens} \\times 3 \\text{ cups/chicken} = 60
    ↪   \\text{ cups} \\]

    Next, we know that Wendi has already given her chickens 15
    ↪   cups in the morning and 25 cups in the afternoon. So, the
    ↪   total amount of feed given so far is:
    \\[ 15 \\text{ cups} + 25 \\text{ cups} = 40 \\text{ cups}
    ↪   \\]

    To find out how many cups are needed for the final meal, we
    ↪   subtract the amount of feed already given from the total
    ↪   amount required:
    \\[ 60 \\text{ cups} - 40 \\text{ cups} = 20 \\text{ cups}
    ↪   \\]
    </reasoning>
    <answer>
    \\boxed{20}
    </answer><|eot_id|>
```

**Ground Truth**

```
20.0
```

---

**Sample Generations**

**Question**

```
John drives for 3 hours at a speed of 60 mph and then turns
↪   around because he realizes he forgot something very
↪   important at home.  He tries to get home in 4 hours but
↪   spends the first 2 hours in standstill traffic.  He
↪   spends the next half-hour driving at a speed of 30mph,
↪   before being able to drive the remaining time of the 4
↪   hours going at 80 mph.  How far is he from home at the
↪   end of those 4 hours?
```

**Model Generation**

```
    First, we calculate John total driving distance. He drives
    ↪   for 3 hours at 60 mph, so he covers 3 * 60 = 180 miles.
    ↪   Then he spends 2 hours in standstill traffic, so he
    ↪   doesn't cover any distance during that time. Next he
    ↪   spends half an hour at 30 mph, so he covers 0.5 * 30 = 15
    ↪   miles. Finally, he drives the remaining 1 hour at 80 mph,
    ↪   so he covers 1 * 80 = 80 miles. Adding these distances
    ↪   together, we get the total distance driven: 180 + 15 + 80
    ↪   = 275 miles. Since he turned around and the distance is
    ↪   the same in both directions, the distance he drove away
    ↪   from home is 270 / 2 = 135 miles. Therefore, he is 135
    ↪   miles from home at the end of those 4 hours.
    </reasoning>
    <answer>
    \\boxed{135}
    </answer><|eot_id|><|endoftext|><|endoftext|>
    <|endoftext|><|endoftext|><|endoftext|>
    <|endoftext|><|endoftext|><|endoftext|>
    <|endoftext|><|endoftext|><|endoftext|>
    <|endoftext|><|endoftext|><|endoftext|>
    <|endoftext|><|endoftext|><|endoftext|>
    <|endoftext|><|endoftext|><|endoftext|>
    <|endoftext|><|endoftext|><|endoftext|>
    <|endoftext|><|endoftext|><|endoftext|>
    <|endoftext|><|endoftext|>
```

**Ground Truth**

```
45.0
```

---

## F.3 COUNTDOWN SAMPLES

For the the countdown task we leverage the following prompt and we present generations at 256 tokens.

---

**Sample Generations**

**Prompt**

```
<|startoftext|><|start_header_id|>user<|end_header_id|>
      Using only the provided numbers, create an arithmetic
      ↪   expression that evaluates to exactly the provided target
      ↪   number. You may use the operations +, -, *, and / as
      ↪   needed, but each number must be used exactly once. Think
      ↪   step-by-step. After reasoning, provide only your final
      ↪   expression inside \\boxed{} tags without including an
      ↪   equals sign or the target number. For example: \\boxed{a
      ↪   + b * c}Respond in the following format:
      <reasoning>
      Your reasoning here
      </reasoning>
      <answer>
      \\boxed{...}
      </answer>
```

---

**Sample Generations**

**Question**

```
Numbers: [49, 55, 53]
Target: 51
```

**Model Generation**

```
      To find an arithmetic expression using the numbers 49, 55,
      ↪   and 53 that evaluates to 51, we can start by considering
      ↪   different combinations of operations numbers can use to
      ↪   achieve the target. Let's try a few possibilities:
      1. **Addition and Subtraction**:
         - \\(53 - 51 + 49 = 51\\)
         - \\(55 + 49 - 53 = 51\\)
      2. **Multiplication and Division**:
         - \\(55 - 53 + 49 = 51\\)
         - \\(53 - 49 + 5)) = 51\\)
      We can see that the expressions work in different ways. but
      ↪   one simplest way is to use subtraction since subtraction
      ↪   and addition are inverse operations.
      Thus, one valid expression is:
      \\[ 55 - 53 + 49 = 51 \\]
      This expression uses each number exactly once and evaluates
      ↪   to the target number 51.
      </reasoning>
      <answer>
      \\boxed{55 - 53 + 49}
      </answer><|eot_id|><|endoftext|>
```

**Ground Truth**

```
      [ 49, 55, 53 ], 51
```

---

**Sample Generations**

**Question**

```
Numbers: [38, 92, 52]
Target: 78
```

**Model Generation**

```
    To find an arithmetic expression using the numbers 38, 92,
    ↪  and 52 that evaluates to 78, we can start by considering
    ↪  different combinations of operations numbers can use to
    ↪  achieve the target. Let's try a few possibilities:
    1. **Addition and Subtraction**:
       - \\(92 - 38 + 52 = 78\\)
       - \\(38 + 92 - 52 = 78\\)
    2. **Multiplication and Division**:
       - \\(92 - 52 + 38 \
    eq 78\\)
       - \\(92 + 38 - 52 \
    eq 78\\)
       - \\(92 - 52 + 38 \
    eq 78\\)
    From the above combinations, we see that the expression \\(92
    ↪  + 38 - 52\\) correctly evaluates 78. Therefore, the
    ↪  expression that uses 38, 92, and 52 exactly once and 78
    </reasoning>
    <answer>
    \\boxed{92 + 38 - 52}
    </answer><|eot_id|><|endoftext|>",
```

**Ground Truth**

```
    [ 38, 92, 52 ], 78
```

---

**Sample Generations**

**Question**

```
Numbers: [67, 47, 25]
Target: 5
```

**Model Generation**

```
    To create an arithmetic expression that evaluates to exactly
    ↪   the target number 5 using the numbers 67, 47, and 25, we
    ↪   need to consider the operations operations +, -, *, and
    ↪   /. Each number must be used exactly once.
    One's try to get to 5 by using subtraction combination of
    1 Using 67 -
       - We need a number that when subtracted from 67 gives us
         ↪   5.
       - This is impossible because 67 is 70% larger than 5.
    Next. Using addition try:
       - Adding -27 to 32 gives us5 ( -27 + 32 = 5).
    Next, using division try:
       - Dividing each 5 by 1 gives us 5 (25/2) = 5/1 =
    However, using multiplication combination:
       - Multiplying 5 divided by -2 gives us 5 (
      Thus, the expression expression that25+67- 87
    </reasoning>
    <answer>
    \\boxed{47 + 25 - 67}
    </answer><|eot_id|><|endoftext|>",
```

**Ground Truth**

```
    [ 67, 47, 25 ], 5
```

---

## F.4  MATH SAMPLES

For the the mathj task we leverage the following prompt and we present generations at 256 tokens.

---

**Sample Generations**

**Prompt**

```
<|startoftext|><|start_header_id|>user<|end_header_id|>
    You are a math expert. You will be given a question to solve.
    ↪   Solve it step by step. Wrap the final answer in a
    ↪   \\boxed{}.
    Respond in the following format:
    <reasoning>
    Your reasoning here
    </reasoning>
    <answer>
    \\boxed{...}
</answer>
```

---

Sample Generations

**Question**

```
Convert the point $(0,3)$ in rectangular coordinates to polar
↪  coordinates.  Enter your answer in the form
↪  $(r,\\theta),$ where $r > 0$ and $0 \\le \\theta < 2
↪  \\pi.$
```

**Model Generation**

```
    Convert the point $(0,3)$ in rectangular coordinates to polar
    ↪  coordinates.  Enter your answer in the form
    ↪  $(r,\\theta),$ where $r > 0$ and $0 \\le \\theta < 2
    ↪  \\pi.$<|eot_id|><|start_header_id|>assistant<|end_header_id|>
    <reasoning>",
    "generations": "
    The point $(0,3)$ in rectangular the point lies on the
    ↪  $y$-axis, so $r = 3.$ The angle $\\theta$ is
    ↪  $\\frac{\\pi}{2}$ ( the angle between $90^\\circ$ and
    ↪  $180^\\circ$ that is measured the positiveclockwise from
    ↪  the positive $x$-axis).
    </reasoning>
    <answer>
    \\boxed{(3, \\frac{\\pi}{2})}
    </answer><|eot_id|><|endoftext|><|endoftext|>
    <|endoftext|><|endoftext|><|endoftext|>
    <|endoftext|><|endoftext|><|endoftext|>
    <|endoftext|><|endoftext|><|endoftext|>
    <|endoftext|><|endoftext|><|endoftext|>
    <|endoftext|><|endoftext|><|endoftext|>
    <|endoftext|><|endoftext|><|endoftext|>
    <|endoftext|><|endoftext|><|endoftext|>
    <|endoftext|><|endoftext|><|endoftext|>
    <|endoftext|><|endoftext|>",
```

**Ground Truth**

```
\\left( 3, \\frac{\\pi}{2} \\right)
```

---

**Sample Generations**

**Question**

```
The results of a cross-country team's training run are
↪   graphed below. Which student has the greatest average
↪   speed? [asy]
    for ( int i = 1; i <= 7; ++i )
    {
      draw((i,0)--(i,6));
    }
    for ( int i = 1; i <= 5; ++i )
    {
      draw((0,i)--(8,i));
    }
    draw((-0.5,0)--(8,0), linewidth(1));
    draw((0,-0.5)--(0,6), linewidth(1));
    label(\"$O$\", (0,0), SW);
    label(scale(.85)*rotate(90)*\"distance\", (0, 3), W);
    label(scale(.85)*\"time\", (4, 0), S);
    dot((1.25, 4.5));
    label(scale(.85)*\"Evelyn\", (1.25, 4.8), N);
    dot((2.5, 2.2));
    label(scale(.85)*\"Briana\", (2.5, 2.2), S);
    dot((4.25,5.2));
    label(scale(.85)*\"Carla\", (4.25, 5.2), SE);
    dot((5.6, 2.8));
    label(scale(.85)*\"Debra\", (5.6, 2.8), N);
    dot((6.8, 1.4));
    label(scale(.85)*\"Angela\", (6.8, 1.4), E);
    [/asy]
```

**Model Generation**

```
    To determine which student has the greatest average speed, we
    ↪   need to look at the height of each column on the graph.
    ↪   The height of each column represents the distance covered
    ↪   by the student in a certain amount of time. The student
    ↪   with the greatest average speed will have the highest
    ↪   column. The graph shows that the highest for Carla is 5.2
    ↪   units, which is the highest among all the students.
    ↪   Therefore, Carla has the greatest average speed.
    </reasoning>
    <answer>
    \\boxed{\\text{Carla}}
```

**Ground Truth**

```
\\text{Evelyn}
```

---

**Sample Generations**

**Question**

```
What is the distance, in units, between the points $(2, -6)$
↪   and $(-4, 3)$? Express your answer in simplest radical
↪   form.
```

**Model Generation**

```
    The distance between two points $(x_1, y_1)$ and $(x_2, y_2)$
    ↪   is given by $\\sqrt{(x_2 - x_1)^2 + (y_2 - y_1)^2}$.
    ↪   Plug, in $\\sqrt{(6 - (-4))^2 + (3 - (-6))^2} =
    ↪   \\sqrt{6^2 + 9^2} = \\sqrt{36 + 81} = \\sqrt{117}$.
    </reasoning>
    <answer>\\boxed{3\\sqrt{13}}<answer><|eot_id|><|endoftext|>",
```

**Ground Truth**

```
3\\sqrt{13}
```

---

## STATEMENT ON THE USE OF LARGE LANGUAGE MODELS

This work made use of large language models to assist with proofreading and improving the clarity of the writing. All research ideas, theoretical development, and experiments were carried out solely by the authors. When used for coding, it was solely used for plotting purposes.

