# OpenReview forum: "Improving Reasoning for Diffusion Language Models via Group Diffusion Policy Optimization"
_ICLR.cc/2026/Conference — ICLR 2026 Poster_

### Official Review · Reviewer_fTLx · 2025-10-31

**Soundness:** 1
**Presentation:** 2
**Contribution:** 2
**Rating:** 4
**Confidence:** 4

**Summary:**

This paper suggests that the direct multiplying token-wise probabilities (used in diffu-GRPO) is not a good estimation of the probability of the whole sequence. Considering the expensive cost strictly following the ELBO, the authors analyze the main cause of the estimation variance, and find that the randomness of sampling $t$ is the main cause. Thus, GDPO uses deterministic time to consider the estimation as a time integral to reduce variance. Experiments indicate that it outperforms the original LLaDA and diffu-GRPO on multiple benchmarks.

**Strengths:**

- This paper provides a detailed analysis on the origin of the variance in diffu-GRPO, taking it as the motivation for deterministic $t$ decisions. This analysis is valuable for DLM RL algorithm designs.
- The comparison between GDPO and diffu-GRPO is comprehensive, including multiple domains (math, planning, coding).
- The paper is well written and easy to follow.

**Weaknesses:**

- I think the most severe problem is that the authors perform too few comparisons, only with LLaDA and diffu-GRPO. Actually, there have been much more related works on DLM RL, such as [1,2]. It is necessary to compare them directly.
- I do not consider diffu-GRPO as a token-level RL method. Actually, it also follows the ELBO in L 195 to provide an approximation for the probability to generate the certain sequence, but with an extremely small number of sample times (N=1). Increasing this could also make the algorithm more stable. We agree that use deterministic $t$ may further reduce the variance. However, other than comparing with diffu-GRPO, the paper does not provide a comparison between this two methods under the same number of MC sample times.
- There also lacks ablation studies to validate the effectiveness of the quadrature rule.

**Questions:**

I think the main weakness is the lack of experiments, see weaknesses for details.

---

> ### Author Response · Authors · 2025-11-26
>
> > I think the most severe problem is that the authors perform too few comparisons, only with LLaDA and diffu-GRPO. Actually, there have been much more related works on DLM RL, such as [1,2]. It is necessary to compare them directly.
>
> We thank the reviewer for noting this. We have included new baselines in our revised manuscript, such as wd1 [1] and coupled-GRPO [2]. For [2], we have also included a detailed comparison in Appendix E (highlighted in $\textcolor{orange}{\textbf{orange}}$). Our results show that GDPO is able to achieve better results than the newly included baselines.
>
> Regarding the methods proposed by the reviewer, it appears that the citations were not included. For this reason, we might've missed them; we hope that our newly added baselines can serve the same purpose.
>
> [1] Tang, Xiaohang, et al. "wd1: Weighted policy optimization for reasoning in diffusion language models." arXiv preprint arXiv:2507.08838 (2025).
>
> [2] Gong, Shansan, et al. "DiffuCoder: Understanding and Improving Masked Diffusion Models for Code Generation." arXiv preprint arXiv:2506.20639 (2025).
>
> > I do not consider diffu-GRPO as a token-level RL method. Actually, it also follows the ELBO in L 195 to provide an approximation for the probability to generate the certain sequence, but with an extremely small number of sample times (N=1). Increasing this could also make the algorithm more stable. We agree that use deterministic further reduce the variance. However, other than comparing with diffu-GRPO, the paper does not provide a comparison between this two methods under the same number of MC sample times.
>
> We must respectfully disagree about diffu-GRPO not being a token-level RL method. Their design and loss function have been specifically tailored to computing the per-token importance ratios.  To this end, they propose a method for approximating them. Specifically, they utilize an approximation of the form:
>
> $$\log p_\theta(y^i|q \oplus M \oplus \dots \oplus M).$$
>
> Where $y^i$ is the $i-$th token of a sequence $y$ with prompt $q$. To evaluate the KL (which requires a sequence-level likelihood) they utilize a mean-field approximation as the product of the **per-token** log probabilities. As the reviewer points out, this mean-field approximation can be understood as approximating the ELBO by using only one sample time.
>
> How to use a larger $N$ to approximate the diffu-GRPO per-token log probabilities has been explored in diffucoder, however, it is unclear how to do this for general $N$ and therefore, we can't perform a comparison against a method that hasn't been described before.
>
> However, in order to do the best possible comparison, we have implemented diffucoder in the countdown and sudoku tasks. Under the same setups that we have used in our paper and presented an in-depth discussion of our differences with diffucoder in Appendix E (highlighted in orange). The results were as following:
>
> | **Model** | **Countdown** |  |  | **Sudoku** |  |  |
> |-----------|:-------------:|:-------------:|:-------------:|:-------------:|:-------------:|:-------------:|
> |           | 128 | 256 | 512 | 128 | 256 | 512 |
> | LLaDA-8B-Instruct | 20.7 | 19.5 | 16.0 | 11.7 | 6.7 | 5.5 |
> | + *diffu-GRPO* | 33.2 | 31.3 | 37.1 | 18.4 | 12.9 | 11.0 |
> | + SFT+*diffu-GRPO* | 34.8 | 32.0 | 42.2 | 22.1 | 16.7 | 9.5 |
> | + coupled-GRPO | 33.59 | 43.75 | 47.66 | 23.63 | 21.0 | 21.92 |
> | + GDPO | **42.97** | **67.19** | **66.41** | **25.05** | **24.17** | **25.10** |
>
> We found that ccoupled-GRPO can significantly improve upon the results of diffu-GRPO, but GDPO obtained the best results, highlighting the strength of our method.
>
> > There also lacks ablation studies to validate the effectiveness of the quadrature rule.
>
> We performed a study on the quality of the quadrature in Figure 3 as well as an ablation on the effectiveness of quadratures with different $N$ when compared to double Monte Carlo approaches in Figure 4. In Figure 3, we found that the quadrature is very effective at approximating the ELBO even at small values of $N$ and that larger values of $N$ might produce only *marginal improvements*. In Figure $4$  we demonstrated that the accuracy of the approximation improves the RL training. We hope that these ablations serve to understand the effectiveness of the quadrature rule.

---

### Official Review · Reviewer_xyfr · 2025-11-01

**Soundness:** 3
**Presentation:** 2
**Contribution:** 3
**Rating:** 6
**Confidence:** 3

**Summary:**

The paper studies the problem reinforcement learning with diffusion language models. A key challenge in RL with DLMs is that likelihoods are hard to estimate. Prior methods either used biased token-level proxies or high-variance sequence-level estimates. As an alternative the authors propose Group Diffusion Policy Optimization (GDPO), which keeps a GRPO-style group advantage but replaces token-level terms with the sequence-level ELBO estimated by a semi-deterministic Monte-Carlo, This consists of fixing a few diffusion timesteps via Gaussian quadrature (2–3 points) and using a single inner MC sample, yielding a provably lower-variance estimator than standard double-MC under tight evaluation budgets. The paper analyzes ELBO variance, showing timestep sampling dominates and  provides asymptotic error bounds/bias discussion for SDMC. Empirically, using LLaDA-8B as the base model, GDPO improves over the base models and over diffu-GRPO across math (GSM8K, MATH500), planning (Countdown, Sudoku), and coding (HumanEval, MBPP), including better length generalization to 512 tokens.

**Strengths:**

* The paper overall is quite well written. The problem is defined quite clearly and approach explained clearly.
* The variance analysis in Section 3.2 is quite useful and clear motivation for the problem.
* The variance reduction approach is explained clearly and coupled with useful theoretical results establishing lower variance. The authors also validate this empirically.
* The resulting algorithm is fairly efficient and simple of implement.
* The empirical results show consistent improvements over some baselines across a variety of different tasks (math, coding, planning).

**Weaknesses:**

* The paper misses some closely related prior work on RL fine-tuning of diffusion language models [1, 2, 3]. In particular, I would like to see a comparison on how the SDMC compares with the variance reduction scheme from [2]. I believe a comparison to these baselines would be critical.
* I also think the empirical results would benefit from having some simple test-time baselines like majority voting.
* The experiments are limited to a single base model. I think it would benefit the paper to have some experiments with another base model (e.g. https://github.com/DreamLM/Dream)


[1] Venkatraman et al., 2024. Amortizing intractable inference in diffusion models for vision, language, and control.

[2] Zhu et al. 2025. LLaDA 1.5: Variance-Reduced Preference Optimization for Large Language Diffusion Models.

[3] Zekri and Boullé, 2025. Fine-Tuning Discrete Diffusion Models with Policy Gradient Methods.

**Questions:**

* Are the results in Table 2 and 3 single seed for training and inference? If so I think it would be better to report mean@k.
* The authors mention that their approach doesn't require too many compute resources, but there is not much detail about how much time training takes on the said resources and how it compares to say diffu-GRPO. Could you please clarify that?

---

> ### Author Response · Authors · 2025-11-26
>
> > The paper misses some closely related prior work on RL fine-tuning of diffusion language models [1, 2, 3]. In particular, I would like to see a comparison on how the SDMC compares with the variance reduction scheme from [2]. I believe a comparison to these baselines would be critical.
>
> This is a very insightful and important question. Our approaches are actually closely related. In LLaDA 1.5, they found that the best way to allocate the Monte Carlo samples is into the time variable, while not using multiple Monte Carlo samples for the inner integral.
>
> Our approach is similar, as we also use only one sample for the inner integral, but further reduce the variance by picking $t$ in a deterministic fashion. In this sense, we have already included the findings from LLaDA 1.5 and further improved upon them.
>
> This comparison between these methods can be found in Figures 3 and  4. In Figure 3, we noticed that our method can better approximate the ELBO than the variance-reduced approach of LLaDA 1.5 and in Figure 4, that our approach also achieves better training on downstream tasks.
>
> > The experiments are limited to a single base model. I think it would benefit the paper to have some experiments with another base model (e.g. https://github.com/DreamLM/Dream)
>
> In order to have comparable results to previous works and fully understand the algorithmic differences we have chosen to use the same base model. Dream is a powerful model that can outperform a tuned LLaDA 8B across several tasks. Furthermore, we would like to highlight that our method is independent of the model, and similar results can be expected.
>
> > Are the results in Table 2 and 3 single seed for training and inference? If so, I think it would be better to report mean@k.
>
> Previous works like [1,2] have used a simple greedy-decoding algorithm, which we also employ during our inference. For this reason, the reported value is mean@1 as opposed mean@k as the accuracy would remain identical across different inference seeds.
>
> As for training, we agree that different seeds and checkpoints can result in significantly different accuracies due to potential large variance issues in RL training. However, following the standard practice we choose to report the performance of our best model to display the maximal capabilities of the algorithm.
>
> [1] Zhao, Siyan, et al. "d1: Scaling reasoning in diffusion large language models via reinforcement learning." arXiv preprint arXiv:2504.12216 (2025).
>
> [2] Tang, Xiaohang, et al. "wd1: Weighted policy optimization for reasoning in diffusion language models." arXiv preprint arXiv:2507.08838 (2025).
>
> > The authors mention that their approach doesn't require too many compute resources, but there is not much detail about how much time training takes on the said resources and how it compares to say diffu-GRPO. Could you please clarify that?
>
>  Although we initially meant this, as in total number of GPUs required to perform the tuning. We agree that understanding the differences in clock time is of great importance, to this end we have set up a simple experiment to collect this data. We set a total clock time of $6$ hours using a per-device batch size of $6$ and compare diffu-grpo with GDPO using 2 or 3 network evaluations per step.
>
> | **Method**            | **Time** | **Number of Iterations** | **NFEs per step**|
> |---------------------|------------------:|------------------:|------------------:|
> | diiffu-GRPO   | 6 | 4500| 1 |
> | GDPO-2     | 6 | 3800 | 2 |
> | GDPO-3 | 6 | 3500 | 3 |
>
> We find that, as expected, increasing the number of network evaluations per step will result in fewer iterations. However, even in our most expensive configuration $N=3$ we still achieve $77$% of the iterations while obtaining the added benefits of GDPO and better likelihood approximation.

---

### Official Review · Reviewer_2CTL · 2025-11-01

**Soundness:** 3
**Presentation:** 3
**Contribution:** 3
**Rating:** 6
**Confidence:** 4

**Summary:**

This paper introduces Group Diffusion Policy Optimization (GDPO), a novel reinforcement learning algorithm tailored to enhance the reasoning capabilities of Diffusion Language Models (DLMs). At its heart, the work tackles a central challenge in applying RL to DLMs: the difficulty of performing tractable likelihood estimation, which has long obstructed conventional policy optimization. The authors pinpoint the high variance and computational burden of existing sequence-level ELBO estimators as the key bottleneck. To overcome this, GDPO pioneers a new, low-variance ELBO estimator built on a Semi-deterministic Monte Carlo scheme—motivated by a fine-grained breakdown of the underlying variance sources. Empirically, GDPO consistently outperforms the strong baseline diffu-GRPO across a variety of reasoning, mathematics, and coding benchmarks, demonstrating its practical strength in fine-tuning DLMs.

**Strengths:**

1. The paper presents a novel idea and introduces a more principled approach for applying GRPO to diffusion large language models. This perspective is insightful and offers valuable inspiration for the research community.
2. GDPO demonstrates substantially better performance than the baseline diffu-GRPO on planning-intensive tasks such as Countdown and Sudoku, highlighting its advantage in structured reasoning and planning.
3. The analysis of variance sources in ELBO approximation is well-motivated, and the proposed semi-deterministic Monte Carlo method appears effective in significantly reducing variance.

**Weaknesses:**

It remains unclear why GDPO underperforms diffu-GRPO on certain mathematical reasoning benchmarks (e.g., GSM8K and MATH 500). The authors are encouraged to provide deeper insight into the potential reasons behind this performance gap.

**Questions:**

1. How was the KL term in GDPO estimated? A more detailed explanation in the paper would be helpful, as this aspect currently seems underexplored.
2. Have the authors experimented with techniques similar to those in GSPO, such as normalizing the reward by the length of the generated sequence?
3. Is the variance reduction technique introduced in this method orthogonal to the approach mentioned in LLaDA 1.5? If so, could they be combined to further enhance performance?

---

> ### Author Response · Authors · 2025-11-26
>
> > It remains unclear why GDPO underperforms diffu-GRPO on certain mathematical reasoning benchmarks (e.g., GSM8K and MATH 500). The authors are encouraged to provide deeper insight into the potential reasons behind this performance gap.
>
> We agree that our improvements relative to D1 are smaller on GSM8K and originally worse in MATH. We would like to note, however, that we didn't use SFT and used only 2 GPUs (4 GPUs for MATH) when compared to the 8 used by D1. Despite this, we achieved better performance on most tasks.
>
> We test the effects of including SFT on the MATH dataset. We trained SFT using the s1k dataset (same as D1) and applied GDPO using $N=3$ (instead of $N=2$ before) with slightly longer completion length 320 (instead of 256 before). We tried one-by-one decoding but didn't adopt it. . We have added these results to the main paper, where we have now been able to beat both D1 and wd1 on the math task. Improving our performance to an impressive $\textbf{\textcolor{green}{impressive  41.4}}$  accuracy at $512$ generation length instead of 38% before. It seems like the benefits of using SFT can be significant and we will continue the exploration of SFT using OpenR1-Math-220K. We thank the reviewer for guiding us in this direction.
>
> Given the result of this experiment, we conclude that the main reason for the original performance gap was the lack of SFT.
>
> > How was the KL term in GDPO estimated? A more detailed explanation in the paper would be helpful, as this aspect currently seems underexplored.
>
> We agree that we didn't explore the use of KL regularization in depth; in fact, in our experiments we did not use any KL regularization following previous works. However, one benefit of GDPO is that it provides a simple way to approximate the KL that we discuss below.
>
> The KL term can be approximated by using the same ELBO approximation as a surrogate for the likelihood. Specifically, we can approximate:
>
> $$\text{KL}(\pi(\cdot \mid q) | p(\cdot \mid q)) = \int \pi(x \mid q) \log \frac{\pi(x \mid q)}{p(x \mid q)}dx \approx \int \mathcal{L}{\text{ELBO}}^{\pi}(x \mid q) \log \frac{ \mathcal{L}{\text{ELBO}}^\pi(x \mid q)}{ \mathcal{L}_{\text{ELBO}}^p(x \mid q)}dx$$
>
>
> One can also use efficient $\text{KL}$ approximators as those used in [1], always replacing likelihoods with ELBO estimates.
>
> [1] Shao, Zhihong, et al. "DeepSeekMath: Pushing the limits of mathematical reasoning in open language models." arXiv preprint arXiv:2402.03300 (2024).
>
>
> > Have the authors experimented with techniques similar to those in GSPO, such as normalizing the reward by the length of the generated sequence?
>
> We thank the reviewer for pointing this out, as displayed in equation 6, we did employ a similar type of normalization. We found that normalizing by the sequence length helped to stabilize the training, similar to what was observed in GSPO.
>
> > Is the variance reduction technique introduced in this method orthogonal to the approach mentioned in LLaDA 1.5? If so, could they be combined to further enhance performance?
>
> This is a very insightful and important question. Our approaches are actually closely related. In LLaDA 1.5 they found that the best way to allocate the Monte Carlo samples is into the time variable, while not using multiple Monte Carlo samples for the inner integral.
>
> Our approach is similar, as we also use only one sample for the inner integral, but further reduce the variance by picking $t$ in a deterministic fashion. In this sense, we have already included the findings from LLaDA 1.5 and further improved upon them.

---

### Official Review · Reviewer_TVg1 · 2025-11-08

**Soundness:** 3
**Presentation:** 2
**Contribution:** 2
**Rating:** 6
**Confidence:** 4

**Summary:**

The paper deal with RLHF with diffusion language models (dLLM). RL methods for fine-tuning LLMs need likelihoods, however DLMs do not provide generation likelihoods. Thus current RL methods either (a) use cheap approximate likelihoods or (b) use the expensive to compute ELBO (which also is approximate but consistent). This work proposes Group Diffusion Policy Optimization (GDPO), which uses ELBO for the likelihood but instead of the. standard Monte-Carlo ELBO, uses low-order quadrature integration to produce a lower variance estimate. Their results show improved performance over both the LLADA and the more recent d1/diffu-GRPO model on a variety of tasks.

**Strengths:**

The problem of inefficient likelihoods for dLLMs is an interesting problem. The semi-deterministic Monte Carlo (SDMC) using quadrature integration is a neat, targeted variance-reduction idea and new to the best of my knowledge.
Experiments are not just limited to more standard tasks like Countdown and GSM used in dLLM literature, but also code related ones.
The paper also provides error analysis for the SDMC estimator, providing insight for determining when their method might give better results.

**Weaknesses:**

Theoretically, the claims rely on smoothness and convex/monotone properties of the integrand tied to KL forms and forward-process assumptions. It’s not fully clear these conditions generally hold.

There are other methods which have been proposed to address the likelihood problem for dLLM tuning. Some of these even focus specifically on variance reduction of the ELBO in dLLMs (for eg diffuCoder). These have not been compared against or even discussed.

The improvements on GSM are minor (and are in fact worse on MATH). They are also worse than wd1, which is a missing baseline on multiple tasks. Similarly coding results are not compared against diffuCoder.

[1] wd1 Weighted Policy Optimization for Reasoning in Diffusion Language Models
[2]  PADRE: Pseudo-Likelihood based Alignment of Diffusion Language Models
[3] Test-Time Alignment of Discrete Diffusion Models with Sequential Monte Carlo
[4] DiffuCoder: Understanding and Improving Masked Diffusion Models for Code Generation

**Questions:**

Some questions have been mentioned in the previous section.Additional to those:

I would like to see some more ablations with varying N (quadrature), not just the accuracy.

 Since the experiments are only with small N (upto 4), what happens if we use a larger N. Since the estimate should get better with higher N, I am also confused why you have small optimal N, instead of as high an N as computationally feasible.

Since there is non-trivial amount of additional computations to compute their approximation to the ELBO, a comparison with baseline models under same computational cost (FLOPS, wall-clock or any other) is important.

Is there some easy metric or other diagnostics to determine if the used N is too small.

---

> ### Author Response · Authors · 2025-11-26
>
> > Theoretically, the claims rely on smoothness and convex/monotone properties of the integrand tied to KL forms and forward-process assumptions. It’s not fully clear these conditions generally hold.
>
> We agree that testing if theoretical conditions hold in practice is a challenging problem. Although we are unable to explicitly test them, judging by our exploratory analysis in section 3.2 it does seem like in practice we have good properties that allow an efficient ELBO estimation. For instance, Figure 2b, shows that the integrand is a monotone function that could be approximated by polynomial functions, since Gauss-Legendre quadratures of order $N$ can approximate integrals of polynomials of order $2N-1$; **if the fine-tuned number of quadrature is large (e.g. 3), that indirectly indicates poor smoothness and a more challenging problem**.
>
> > There are other methods which have been proposed to address the likelihood problem for dLLM tuning. Some of these even focus specifically on variance reduction of the ELBO in dLLMs (for eg diffuCoder). These have not been compared against or even discussed.
>
> To the best of our knowledge, the other work that explored variance reduction for ELBO approximation is LLada 1.5 [2], which corresponds to doing a double Monte Carlo approximation by using all the sampling budget on $t$, which is our main baseline in section 3.3 and in Figure 4.
>
> Although diffucoder [1] does explore a variance reduction technique, it does so for a token-level likelihood, not to approximate the ELBO. They achieve this by using complementary masks as explained in their Appendix A.4. Since we approximate different objectives, the methods are not comparable as a likelihood estimation technique. However, we agree that diffucoder is a similar method to ours, so we highlight some of the differences.
>
> Diffucoder's coupled-GRPO uses $2$ times with complementary masks and is specifically tailored towards their problem. On the other hand, we present a general approach to ELBO approximation where an arbitrary number of time points $N$ can be used; furthermore, our method does not rely on complementary masks. The coupled-GRPO approach works well when the discretization error between time steps is not too large, but it can result in lower accuracy when lots of correlations between the tokens exist. On the other hand, our method can handle this case by increasing $N$.
>
> To further understand the algorithmic differences, we implement coupled-GRPO as a baseline on the sudoku and countdown tasks. We report the comparison here as well as a detailed discussion on Appendix E (highlighted in orange).
>
>
> | **Model** | **Countdown** |  |  | **Sudoku** |  |  |
> |-----------|:-------------:|:-------------:|:-------------:|:-------------:|:-------------:|:-------------:|
> |           | 128 | 256 | 512 | 128 | 256 | 512 |
> | LLaDA-8B-Instruct | 20.7 | 19.5 | 16.0 | 11.7 | 6.7 | 5.5 |
> | + *diffu-GRPO* | 33.2 | 31.3 | 37.1 | 18.4 | 12.9 | 11.0 |
> | + SFT+*diffu-GRPO* | 34.8 | 32.0 | 42.2 | 22.1 | 16.7 | 9.5 |
> | + coupled-GRPO | 33.59 | 43.75 | 47.66 | 23.63 | 21.0 | 21.92 |
> | + GDPO | **42.97** | **67.19** | **66.41** | **25.05** | **24.17** | **25.10** |
>
> We observe that coupled-GRPO can significantly improve the results when compared to diffu-GRPO, but GDPO consistently outperforms coupled-GRPO, demonstrating the strength of our method.
>
> [1] Gong, Shansan, et al. "DiffuCoder: Understanding and Improving Masked Diffusion Models for Code Generation." arXiv preprint arXiv:2506.20639 (2025).
>
> [2] Zhu, Fengqi, et al. "LLaDA 1.5: Variance-Reduced Preference Optimization for Large Language Diffusion Models." arXiv preprint arXiv:2505.19223 (2025).

---

> ### Author Response · Authors · 2025-11-26
>
> > The improvements on GSM are minor (and are in fact worse on MATH). They are also worse than wd1, which is a missing baseline on multiple tasks. Similarly coding results are not compared against diffuCoder.
>
> We agree that our improvements relative to D1 are smaller on GSM8K and originally worse in MATH. We would like to note, however, that we didn't use SFT and used only 2 GPUs (4 GPUs for MATH) when compared to the 8 used by D1 , despite this we achieved better performance on most tasks.
>
> We test the effects of including SFT on the MATH dataset. We trained SFT using the s1k dataset (same as D1) and applied GDPO using $N=3$ (instead of $N=2$ before), one-by-one decoding with slightly longer completion length 320 (instead of 256 before). We have added these results to the main paper, where we have now been able to beat both D1 and wd1 on the math task. Improving our performance to an $\textbf{\textcolor{green}{impressive  41.4}}$ accuracy (instead of 38) at $512$ generation length. It seems like the benefits of using SFT can be significant and we will continue the exploration of SFT using OpenR1-Math-220K. We thank the reviewer for guiding us in this direction.
>
> Regarding other benchmarks, we thank the reviewer for noting them. We must admit that we were unaware of some of these, as they are very recent; we have included citations to these works and, when appropriate, included them in the results table.
>
> Regarding comparison with diffu-coder on coding tasks, we didn't choose to make this comparison as the settings are extremely different. Diffu-coder is a large scale training of a specialized diffusion language models for coding, they train over 700B tokens using 8 to 10 nodes of 8H100s compared to our much smaller experiment using a more modest 8GPUs on the KodCode-Light-RL with 10K samples.
>
> Given the enormous difference in scale we find that it is not a proper comparison and doesn't highlight the differences in the algorithm. However, we agree that having a comparison against diffu-coder's coupled-GRPO is of great importance, so we hope that the experiment above as well as our discussion in **Appendix E** (evaluated on Countdown and Sudoku) helps to close this gap.
>
> > Since the experiments are only with small N (upto 4), what happens if we use a larger N. Since the estimate should get better with higher N, I am also confused why you have small optimal N, instead of as high an N as computationally feasible.
>
> Thanks for this insightful comment. As indicated by the result of Figure 3, using a large N leads to only marginal gains in the accuracy of the approximation. Therefore, we empirically adopt $N=2$ as our default and use $N=3$ when the compute allows it. On the other hand, using large N is a challenging computation problem, as we need to backpropagate through each evaluation. Large N means that the computational graph grows very fast, limiting the ability to train the model.
>
> > Is there some easy metric or other diagnostics to determine if the used N is too small.
>
> We recommend testing the accuracy of the ELBO estimation against a Double Monte Carlo approximation. This procedure is illustrated in Figure 3, where we observed that a larger N didn't significantly increase the accuracy. Generally, the shape of this curve conveys most of the information. Another such diagnostic is leveraging numerical analysis theory, which tells us that Gaussian quadratures of order $N$ can accurately approximate polynomials of degree $2N-1$. As observed by plotting the loss function in Figure 2b, the integrand has a simple shape that could be approximated by a low-order polynomial, allowing us to choose a smaller $N$.

---

### Author Response · Authors · 2025-11-24

Dear Reviewers,

Thanks for your patience. Adding baselines like DiffuCoder, refining the Math performance, and running some ablation studies required some additional time, but everything is progressing well. We will update the rebuttal shortly.

Anonymous Authors

---

### Comment · Area_Chair_nwRH · 2025-11-28
**paper is deanonymised**

Dear authors,

The updated PDF is not anonymised.

This has unfortunately been noticed by at least some reviewers, and I’m afraid it will lead to automatic rejection.

AC

---

> ### Author Response · Authors · 2025-11-28
> **apologies for the oversight**
>
> In preparation for the final submission, it seems there was an oversight. This author is trying to find out what happened. I have removed the PDF and update when other authors are awake.

---

### Author Response · Authors · 2025-11-28

We thank the reviewers for their time and helpful comments that have allowed us to improve our submission. After the revision we have incorporated the following changes:

1. Improved the performance of the MATH dataset by including SFT, one-by-one decoding, and using $3$ quadrature points instead of $2$
2. Implemented coupled-GRPO (diffucoder) on the sudoku and countdown benchmarks to provide a better comparison with our method as well as a detailed comparison of the difference between GDPO and coupled-GRPO in appendix E
3. Added a detailed analysis of the runtime of GDPO in Appendix D.4
4. Improved figure 3 to better highlight the fast convergence of Gaussian quadratures for ELBO approximation
5. Included new citations to relevant works and more baselines (including wd1 and coupled-GPRO) as suggested by the reviewers

We hope that these new additions can improve the quality of our submission and are grateful for the contructive criticism. We are also very sorry about the de-anonymization, which we believe has been rectified as fast as possible, and we truly hope its impact could be minimized and our paper still be evaluated objectively based on scientific contribution. Your understanding would be deeply appreciated especially given the turmoil.


Sincerely,

The Authors

---

### Author Response · Authors · 2025-11-30

Dear Reviewers,

We have further improved the best performance of the gsm8k dataset from 82.2% to 85.0% by including SFT, using 3 quadrature points, one-by-one decoding, and 320 completion length. We have added these changes to the revised version.

Kind regards,

The authors

---

### Author Response · Authors · 2025-12-02

Dear Reviewers,

We are thankful for your time and helpful comments that have allowed us to improve our submission. Following the suggestions provided during the rebuttal phase, we incorporated supervised fine-tuning (SFT), increased the number of quadrature points to 3, and extended the completion length to 320 tokens during training. During inference, we adopted one-by-one decoding instead of parallel decoding. Below we outline the improvement of our best model in each task:

- GSM8K: 82.26 -> 84.99
- MATH500: 38.2 -> 41.4
- Sudoku: 25.10 -> 27.69
- Countdown: 67.19 -> 80.86

We have updated the submission to reflect the new results, which make our results comparable to those reported by frontier research labs [1]. In the final version, we will conduct a comprehensive ablation study to isolate each of these components.

We are very grateful for the helpful comments that led to these improvements!

Sincerely,

The Authors

[1] "Spg: Sandwiched policy gradient for masked diffusion language models." arXiv preprint arXiv:2510.09541 (2025).

---

### Meta-Review · Area_Chair_3G7h · 2025-12-26

**Summary:**

This paper proposes a new method GDPO, which improves RL training of diffusion language models (DLMs) by a semi-deterministic Monte Carlo estimation method for the sequence-level ELBO.

Empirical results show that GDPO achieves consistent gains over pretrained DLMs and outperforms previous methods, which is also supported by theoretical results. These results are important since it mitigates one of the key challenges of likelihood evaluation in RL for DLM.

Two major concerns from the reviewers in the orignial reviewers are

- Missing baselines
- Underperformance on Math benchmarks

The two concerns which have been addressed by the authors through additional experiments in revision. Therefore, I think the paper after its revision and rebuttal phase, will make a significant contribution to the DLM community and I would like to recommend an acceptance.

**Reviewer Concerns:**

Most concerns from the reviewers have been addressed by the authors' rebuttal and paper revisions. However, there remain some space for the study to further improve:
- Single base model
- Theoretical assumptions (Smoothness & Convexity): more evidence (both theoretical and empirical, such as using more base models) could support the authors' claims.
- Single training seed: the authors are encouraged to report more random seeds in the next version

**Reviewer Scores:**

I expect Reviewer fTLx to increase his/her score from 4 to 6, and the other 3 reviewers are also likely to support acceptance with higher confidence.

---

### Decision · Program_Chairs · 2026-01-26

Accept (Poster)